# Simulating Mixed State Dynamics to Enable Differentiable Quantum Architecture Search

## Abstract

Variational Quantum Algorithms (VQAs) are a promising approach to leverage Noisy Intermediate-Scale Quantum (NISQ) computers. However, choosing optimal quantum circuits that efficiently solve a given VQA problem is a non-trivial task. Quantum Architecture Search (QAS) algorithms enable automatic generation of quantum circuits tailored to the provided problem. Existing QAS approaches typically adapt classical neural architecture search techniques, training machine learning models to sample relevant circuits, but often overlook the inherent quantum nature of the circuits they produce. By reformulating QAS from a quantum perspective, we propose a sampling-free differentiable QAS algorithm that models the search process as the evolution of a quantum mixed state, which emerges from the search space of quantum circuits. The mixed state formulation also enables our method to incorporate generic noise models, for example the depolarizing channel, which cannot be modeled by state vector simulation. We validate our method by finding circuits for state initialization and Hamiltonian optimization tasks, namely the variational quantum eigensolver and the unweighted max-cut problems. We show our approach to be comparable to, if not outperform, existing QAS techniques while requiring significantly fewer quantum simulations during training, and also show improved robustness levels to noise.

## 1 Introduction

Variational Quantum Algorithms (VQAs) leverage hybrid quantum–classical optimization and have proved to be a crucial mainstay for securing Noisy Intermediate-Scale Quantum (NISQ) computers quantum advantage. Noteworthy instances of this leverage include Variational Quantum Eigensolvers (VQEs) for chemical simulations (Tilly et al., 2022) and the Quantum Approximate Optimization Algorithm (Farhi et al., 2014) for solving combinatorial optimization problems. Moreover, when Parameterized Quantum Circuits (PQCs) are applied to machine learning tasks, VQAs can be interpreted as Quantum Neural Networks (QNNs). Recent research has demonstrated that QNNs can be strictly more expressive than comparably-sized classical networks (Du et al., 2020) with empirical evidence also illustrating that QNNs match or exceeded the performance of classical networks, while using much fewer parameters (Bischof et al., 2025). These results highlight that properly structured QNNs can capture subtleties of complex functions more efficiently than classical networks.

Determining optimal PQCs is a difficult task in this NISQ era as circuits need to solve the underlying tasks while also being resilient to noise. Quantum Architecture Search (QAS) algorithms provide a way to automate the design of PQCs. A wide range of QAS strategies, including reinforcement learning, evolutionary algorithms and generative models, have been proposed to address the challenges of quantum circuit design (Martyniuk et al., 2024). Among these, we are particularly interested in differentiable QAS algorithms, which allow for gradient-based optimization, and are largely inspired by the classical differentiable neural architecture search, DARTS (Liu et al., 2018).

Within the DARTS framework, neural network architectures are modeled as a sequence of operations taken from a candidate operation set. To make a task differentiable and solvable using gradient descent, the discrete search space of architectures is relaxed to a continuous domain of parameters, giving rise to probability distributions that identify operations for each position in the sequence. Therefore, the output of each block is formulated as the sum over the outputs for each operation, weighted by the softmax probabilities of selecting the operations. In this manner, the architecture

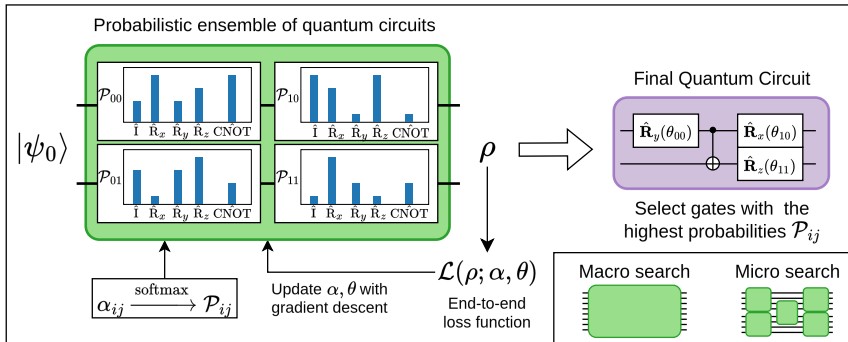

Figure 1: A schematic overview of $\rho$DARTS showing the optimization loop. $\rho$DARTS provides for macro and micro searches which generate global circuits and local subcircuits, respectively.

and the parameters for each operation are trainable end-to-end to minimize the overall task loss. After training, operations with the highest probabilities will form the final neural network architecture.

While DARTS has proven valuable in classical neural networks, a major barrier in extending classical DARTS to differentiable QAS algorithms lies in the representation of the quantum state itself. In contrast to classical computing, where the softmax weighted sum of states remains valid, an arbitrary sum of quantum state vectors weighted by probabilities does not yield a valid quantum state. For this reason, differentiable QAS algorithms approximate the gradients of the architecture parameters by sampling circuits from the search space and evaluating the quantum states they generate (Zhang et al., 2022; Wu et al., 2023). However, since the architecture gradients are only influenced by the sampled circuits, these QAS approaches are restricted to a limited view of the full search space.

We propose to resolve this inherent limitation by considering a quantum mixed state constructed from the entire search space of quantum circuits. In contrast to state vectors, mixed states, represented by density matrices, guarantee that a probability-weighted sum of quantum states produces a valid mixed state (Nielsen & Chuang, 2010). Furthermore, density matrix representations enable the consideration of general noise models, including the depolarizing channel, which cannot be modeled with state vectors.

Therefore, we introduce $\rho$DARTS as a new differentiable QAS algorithm that models the search process using density matrix simulations, see Fig. 1. Our algorithm models the architecture search space as a probabilistic ensemble of quantum architectures, which naturally gives rise to mixed states. In particular, we make the following contributions:

- A sampling-free, differentiable QAS approach based on the dynamics of mixed states, translating the classical DARTS algorithm to a quantum-native setting.
- A framework to model realistic quantum noise channels during the search process.
- Extensive experiments that demonstrate the efficacy of our algorithm to find PQCs for state initialization, VQE and unweighted max-cut problems.

## 2 RELATED WORK

Within the quantum machine learning community, large attraction has been drawn towards PQCs, which directly compete with classical neural networks and promise improved parameter efficiency. However, compared to deep learning, where best practices for architectures where found early on and continuously improved (Simonyan & Zisserman, 2015; He et al., 2016; Dosovitskiy et al., 2021), the design space of PQCs is considerably larger, while at the same time, execution and simulation are limited by today's hardware and exponential growth of computational cost on classical systems. Approaching this challenge, QAS algorithms automate the PQC design process by formulating the task as either differentiable or non-differentiable optimization problems.

**Non-differentiable QAS algorithms** QAS algorithms automate the design of PQCs for variational quantum algorithms. A broad range of QAS strategies have been proposed to address the challenges of manual circuit design, including hardware constraints and noise sensitivity (Martyniuk et al.,

2024). The first group of approaches directly considers the non-differentiable nature of the problem. Reinforcement learning approaches to QAS involve learning optimal strategies to construct quantum circuits incrementally (Ostaszewski et al., 2021; Dai et al., 2024; Patel et al., 2024; Kundu & Sarra, 2024; Olle et al., 2025). Quantum Noise-Adaptive Search is an evolutionary algorithm that searches for the subcircuits of a predefined super circuit architecture under a realistic noise model (Wang et al., 2022). Adaptive methods, such as ADAPT-VQE (Grimsley et al., 2019), iteratively build quantum circuits by making use of operators that maximize a performance gradient, which allows for task-specific, compact circuit designs. The Generative Quantum Eigensolver employs a generative pre-trained transformer to produce quantum architectures for VQE tasks (Nakaji et al., 2024). TF-QAS (He et al., 2024) is a training-free QAS method that randomly selects circuits, ranking them according to their topological complexity and expressivity. Another common approach to QAS involves pruning gates from a large PQC to remove redundant gate structures that contribute to barren plateaus (Sim et al., 2021; Hu et al., 2022; Imamura et al., 2023).

**Differentiable QAS algorithms** Other notable methods relax the discrete structure of QAS to a continuous domain, which enables gradient-based optimization, and takes inspiration from classical differentiable neural architecture search, DARTS (Liu et al., 2018). Differentiable Quantum Architecture Search (DQAS) (Zhang et al., 2022) and QuantumDARTS (qDARTS) (Wu et al., 2023) define QAS within a shared framework: circuits are constructed as sequences of unitary operations, selected from a predefined gate set, and the search process is a differentiable bi-level optimization problem. The discrete search space of circuits is relaxed to a learnable probabilistic model, allowing gradient-based updates of both architectural and gate parameters. DQAS (Zhang et al., 2022) updates its architecture parameters using Monte Carlo gradients, which are obtained by sampling a fixed number of circuits in each iteration. By contrast, qDARTS uses Gumbel-softmax reparameterization (Jang et al., 2016) to enable differentiable sampling, optimizing the shared gate parameters for each sampled circuit in an inner loop before updating the architecture parameters.

## 3 PRELIMINARIES

Quantum machine learning executes computations on a system represented by a quantum state, rather than a classical state. This enables strong speedups on specific sets of problems by utilizing properties such as entanglement, superposition, and the probabilistic nature of the state itself. An introduction to the relevant basics of quantum computing and VQAs is provided in appendix B, and the representation of mixed states, the core formalism of our method, is provided in the following.

**Mixed states** Consider a system where the exact quantum state is unknown, but is modeled as the state $|\psi_i\rangle = [\psi_{i,0} \quad \psi_{i,1} \quad \cdots \quad \psi_{i,N-1}]^{\mathsf{T}}$ with probability $p_i$. This *mixed state* is represented by the density matrix $\rho = \sum_i p_i |\psi_i\rangle \langle\psi_i|$, where $\langle\psi_i| = |\psi_i\rangle^{\dagger}$, with $\dagger$ denoting the complex conjugate transpose. Furthermore, evolving a mixed state involves applying a unitary operator, $\hat{U}$, such that $\rho_{t+1} = \hat{U}\rho_t\hat{U}^{\dagger}$. The measurement probabilities of a mixed state $\rho$ are encoded in its diagonal entries; $\rho_{kk}$ is the probability of measuring the qubits as binary representation of the integer $k$.

Density matrices provide a convenient framework for modeling classical uncertainties of quantum states, and are often employed to model noisy quantum systems. Common noise models include the bit flip, phase flip, and depolarizing channels:

$$\text{BitFlip}(\rho, p) = (1-p)\,\rho + p\,\hat{X}\rho\hat{X}^{\dagger}, \quad \text{PhaseFlip}(\rho, p) = (1-p)\,\rho + p\,\hat{Z}\rho\hat{Z}^{\dagger},$$
$$\text{Depolarizing}(\rho, p) = (1-p)\,\rho + \tfrac{p}{N}\hat{I}_N. \tag{1}$$

## 4 METHOD

The goal of $\rho$DARTS is to find an optimal quantum circuit for a given VQA task constructed using gates from candidate gate set, $\mathcal{G}$. The VQA's loss function, $\mathcal{L}$, is naturally extended to the mixed state formalism, allowing the optimal architecture, $\mathcal{A}^*$, and its underlying gate parameters, $\theta$, to be found by minimizing $\mathcal{L}$ evaluated on a mixture of the architecture search space.

We adopt the search space, $\mathcal{S} = \{0, \ldots, |\mathcal{G}| - 1\}^{m \times n}$, specified in qDARTS as an $n$-qubit architecture with $m$ layers is represented by a matrix $\mathcal{A} \in \mathcal{S}$. For gate parameters $\theta \in \mathbb{R}^{m \times n}$, $\mathcal{A}$ defines the

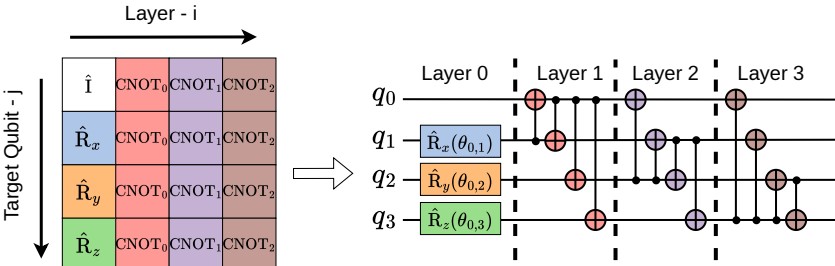

Figure 2: A matrix encoding of a 4-qubit circuit comprising all of the gates in our chosen gate set, and the associated quantum circuit.

quantum circuit

$$\hat{\mathbf{U}}_{\mathcal{A}}(\theta) = \prod_{i=0}^{m-1} \prod_{j=0}^{n-1} \hat{\mathcal{G}}_{\mathcal{A}_{ij}}(\theta_{ij}). \tag{2}$$

The architecture parameters, $\alpha \in \mathbb{R}^{m \times n \times |\mathcal{G}|}$, define softmax probability distributions for selecting a gate at each position in the circuit from the candidate gate set, $\mathcal{G}$. In particular, we have

$$\mathcal{P}_{ij}^{(k)} = \Pr\left(\mathcal{A}_{ij} = k\right) = \frac{\exp\left(\alpha_{ij}^{(k)}\right)}{\sum_{k'=0}^{|\mathcal{G}|-1} \exp\left(\alpha_{ij}^{(k')}\right)}, \tag{3}$$

following which, the probability of selecting an architecture $\mathcal{A}$ is $\mathcal{P}_{\mathcal{A}} = \prod_{i=0}^{m-1} \prod_{j=0}^{n-1} \mathcal{P}_{ij}^{(\mathcal{A}_{ij})}$.

We define quantum operations $\mathcal{E}_{ij}$ that apply the ensemble of gates at layer $i$ to the target qubit $j$,

$$\mathcal{E}_{ij}(\rho) = \sum_{k=0}^{|\mathcal{G}|-1} \mathcal{P}_{ij}^{(k)} \, \hat{\mathcal{G}}_k(\theta_{ij}) \, \rho \, \hat{\mathcal{G}}_k(\theta_{ij})^{\dagger}. \tag{4}$$

By successively applying $\mathcal{E}_{ij}$s on each position in the circuit, we generate a density matrix representing a mixture of output states of every circuit in the search space, which is given by

$$\rho' = \mathcal{E}_{m-1,n-1}(\cdots(\mathcal{E}_{00}(\rho))\cdots) = \sum_{\mathcal{A} \in \mathcal{S}} \mathcal{P}_{\mathcal{A}} \hat{\mathbf{U}}_{\mathcal{A}}(\theta)\rho\hat{\mathbf{U}}_{\mathcal{A}}(\theta)^{\dagger}. \tag{5}$$

Note that since $\rho'$ represents a mixed state over the entire search space, and since the operations $\mathcal{E}_{ij}$ are differentiable, the density matrix simulation allows training without sampling random circuits from the search space. Moreover, the loss function, $\mathcal{L}(\rho; \alpha, \theta)$, can be used simultaneously to train both the architecture and the gate parameters end-to-end.

## 4.1 SEARCH SETTINGS

We define two different search settings for $\rho$DARTS. Algorithms 1 and 2 describe our approach for a macro search and micro search, respectively. The macro search setting computes an optimal architecture for the entire circuit without assuming a predefined circuit structure. On the other hand, the micro search setting is better suited for VQAs, when the circuit structure may be inferred from the optimization problem. This feature is similar to the classical DARTS setting in which the entire architecture is either optimized or sections are repeated throughout the architecture.

Our micro search algorithm learns the architecture of a subcircuit acting on a subset of the qubits present in the system. Multiple copies of this subcircuit are then combined to form the final circuit, following a predefined circuit structure $C \in \{0, \cdots, n-1\}^{N_c \times n_s}$, with $n_s$ denoting the number of qubits each subcircuit acts on, and $N_c$ representing the number of subcircuits. Each copy of the subcircuit possesses its own parameter set $\theta^{(i)}$. This approach reduces the search space to $\mathcal{S} = \{0, \cdots, |\mathcal{G}| - 1\}^{m \times n_s}$, enabling a more amenable optimization. In this setting, we have a sequence

**Algorithm 1** $\rho$DARTS macro search

**Require:** number of qubits $n$, number of layers $m$, candidate gate set $\mathcal{G}$, randomly initialized $\alpha$ and $\theta$, initial state $|\psi_0\rangle$, $num\_epochs$
  **for** $epoch \leftarrow 1$ to $num\_epochs$ **do**
    $\rho \leftarrow |\psi_0\rangle \langle\psi_0|$
    **for** $i \leftarrow 0$ to $m - 1$ **do**
      **for** $j \leftarrow 0$ to $n - 1$ **do**
        $\rho \leftarrow \mathcal{E}_{ij}(\rho)$
      **end for**
    **end for**
    Calculate loss $\mathcal{L}(\rho; \alpha, \theta)$
    Update $\alpha, \theta$ by gradient descent
  **end for**
Fix the final circuit architecture $\mathcal{A}^* \in \mathcal{S}$ such that $\mathcal{A}_{ij}^* = \arg\max_k \mathcal{P}_{ij}^{(k)}$

**Algorithm 2** $\rho$DARTS micro search

**Require:** number of qubits $n$, number of qubits in each subcircuit $n_s$, number of layers $m$, number of subcircuits $N_c$, candidate gate set $\mathcal{G}$, super circuit structure $C$, randomly initialized $\alpha$ and $\theta$, initial state $|\psi_0\rangle$, $num\_epochs$
  **for** $epoch \leftarrow 1$ to $num\_epochs$ **do**
    $\rho \leftarrow |\psi_0\rangle \langle\psi_0|$
    **for** $c \leftarrow 0$ to $N_c - 1$ **do**
      $q \leftarrow C[c, :]$
      **for** $i \leftarrow 0$ to $m - 1$ **do**
        **for** $j \leftarrow 0$ to $n_s - 1$ **do**
          $\rho \leftarrow \mathcal{E}_{ij}^{(q,c)}(\rho)$
        **end for**
      **end for**
    **end for**
    Calculate loss $\mathcal{L}(\rho; \alpha, \theta)$
    Update $\alpha, \theta$ by gradient descent
  **end for**
Fix the final subcircuit architecture $\mathcal{A}^* \in \mathcal{S}$ such that $\mathcal{A}_{ij}^* = \arg\max_k \mathcal{P}_{ij}^{(k)}$

of quantum operations given by

$$\mathcal{E}_{ij}^{(q,c)}(\rho) = \sum_{k=0}^{|\mathcal{G}|-1} \mathcal{P}_{ij}^{(k)} \, \hat{\mathcal{G}}_k^{(q_j)}(\theta_{ij}^{(c)}) \, \rho \, \hat{\mathcal{G}}_k^{(q_j)}(\theta_{ij}^{(c)})^\dagger, \tag{6}$$

where $q$ is a subset of $n_s$ qubit indices, $c$ is the index of the subcircuit, and $\hat{\mathcal{G}}_k^{(q_j)}$ is the gate $\hat{\mathcal{G}}_k$ applied to the $q_j$-th qubit. The computational complexity of our method is discussed in appendix D.

**Entropy regularization** The probability distributions, $\mathcal{P}_{ij}$, identify which gates are applied at a particular position in order to ensure the optimal architecture. The entropy of each distribution is given by $S_{ij} = -\sum_{k=0}^{|\mathcal{G}|-1} \mathcal{P}_{ij}^{(k)} \ln \mathcal{P}_{ij}^{(k)}$, which quantifies the uncertainty of those gates that should be present in the final architecture. During the course of the training, a high-entropy distribution implies that search space is under exploration, while a low-entropy distribution implies that a suitable gate has been selected. Should all distributions $\mathcal{P}_{ij}$ record low entropies, we can infer that the search has converged to a single architecture.

To control the state of exploration, we introduce a regularization term, which we have based on the normalized mean entropy (NME) of the gate distributions,

$$\text{NME}(\alpha) = \frac{1}{mn} \sum_{i=0}^{m-1} \sum_{j=0}^{n-1} \frac{S_{ij}}{\ln |\mathcal{G}|}. \tag{7}$$

In particular, we have $\text{EntropySchedule}(\alpha, t) = S_E(t) \cdot \text{NME}(\alpha)$, where $S_E(t)$ is a scheduler function that interpolates from a minimum value $s_0$ to a positive, maximum value $s_1$ over time. If $s_0$ is negative, the algorithm is driven to explore the full search space at the beginning of the search. Otherwise, the algorithm immediately penalizes high-entropy distributions. In our experiments, we used a sinusoidal interpolation that reached the max value after half of the total epochs,

$$S_E(t) = \begin{cases} s_0 + (s_1 - s_0)\sin(\pi t) & \text{for } 0 \leq t \leq 0.5, \\ s_1 & \text{for } 0.5 < t \leq 1 \end{cases}. \tag{8}$$

**Angle regularization** To limit the redundancy of gate parameters $\theta$, we introduce a differentiable regularization term to penalize any rotation angles outside the range $[-\pi, \pi]$,

$$\text{AnglePenalty}(\theta) = s_\theta \sum_{i,j} \left(\text{ReLU}(\theta_{ij} - \pi) + \text{ReLU}(-\theta_{ij} - \pi)\right)^2. \tag{9}$$

## 5 EXPERIMENTS

$\rho$DARTS is implemented in PyTorch (Imambi et al., 2021) with custom GPU kernels written using the Numba CUDA JIT compiler (Lam et al., 2015). We implemented qDARTS, according to the authors' specifications, to serve as a benchmark for our QAS experiments, for more details see appendix C. In each task, we ran qDARTS and $\rho$DARTS with identical hyperparameters, see appendix E for the hyperparameter values used. We employed the Adam optimizer (Kingma & Ba, 2014) with a cosine annealing learning rate scheduler (Loshchilov & Hutter, 2016) to update the architecture and gate parameters.

**Gate set** The gate set $\mathcal{G}$ we chose for our experiments contains the gates $\hat{I}, \hat{R}_x, \hat{R}_y, \hat{R}_z$, and CNOT. Note that for a search space over $n$ qubits, there are $n-1$ CNOT gates with a specific target qubit, as visualized in Fig. 2. The gate set was chosen since single-qubit gates along with CNOT gates form a universal quantum gate set and all single qubit gates can be decomposed into a sequence of Pauli rotations gates, up to a global phase (Nielsen & Chuang, 2010; Williams, 2011).

**Ablation** To increase the number of trainable parameters associated with the architecture search, we adopt the ablation from qDARTS where, given a number of *hidden units* $K$, each $\alpha_{ij}$ is computed as the product of hidden matrix, $\mathbf{H}_{ij} \in \mathbb{R}^{|\mathcal{G}| \times K}$, and hidden vector, $\vec{v}_{ij} \in \mathbb{R}^K$. In our experiments, we chose the number of hidden units to be $K = 2|\mathcal{G}|$.

### 5.1 TASK I: STATE INITIALIZATION

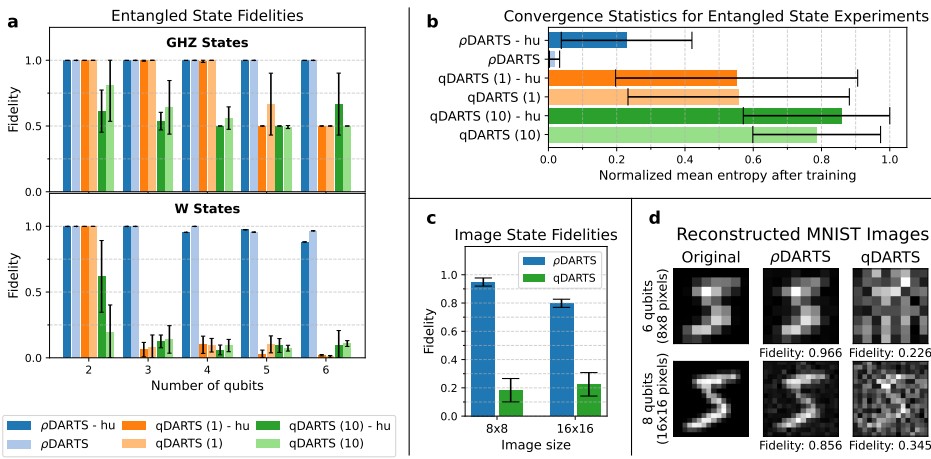

Figure 3: Results of state initialization experiments. a) The fidelities of the states found by $\rho$DARTS and qDARTS, using 1 and 10 quantum simulations per epoch, when initializing the GHZ and W states, averaged over three runs. b) Comparing the normalized mean entropies of the gate distributions after every search for the entangled state experiments. c) The average fidelities of the states found when initializing the image states. d) The best image states found for the digit 5. The error bars indicate one standard deviation from the mean, clipped between 0 and 1.

**Background** Entangled states are multi-qubit states that exhibit strong correlations among the measurement outcomes of their qubits, and as a result, cannot be factored as a product of single qubit states. Examples include the Bell states $|\Phi\rangle = \frac{1}{\sqrt{2}}(|00\rangle + |11\rangle)$ and $|\Psi\rangle = \frac{1}{\sqrt{2}}(|01\rangle + |10\rangle)$, which are two-qubit entangled states. Further, $|\Phi\rangle$ generalizes to the $n$-qubit GHZ state, which, when measured, returns an identical bit value for all qubits, and $|\Psi\rangle$ generalizes to the $n$-qubit W state, whose measurements return a single bit value 1 among all of the qubits.

The circuits that generate these entangled states are known, see appendix G, and they serve as a benchmark to test our QAS methodology. However, in a practical setting, QAS would be used for state initialization when an efficient circuit is not known apriori. For this purpose, we also consider experiments to initialize amplitude-encoded images from the MNIST dataset (Deng, 2012).

The protocol to convert images to amplitude-encoded quantum states and vice versa is described in appendix G.

**Experiment setting** We employed our algorithm to search for circuits that map the initial state $|0\cdots0\rangle$ to the $n$-qubit GHZ and W states using our macro search algorithm with $m = 3$ and $m = 3(n-1)$ layers for the GHZ and W circuits respectively. These layer counts were determined from the known state preparation circuits, see appendix G for more details. We also searched for circuits to prepare one amplitude-encoded image for each handwritten digit in the MNIST dataset. The images were downscaled from the original 28x28 pixels to 8x8 pixels (6 qubits) and 16x16 pixels (8 qubits), using 18 and 96 layers in the macro search setting, respectively.

Fidelity, $\mathcal{F}$, was used to measure the closeness between the state prepared by our search algorithm, $\rho$, and the reference state, $|\phi\rangle$. In particular, we have $\mathcal{F}(\rho, |\phi\rangle) = \langle\phi| \rho |\phi\rangle$, giving the loss function

$$\mathcal{L}(\rho; \alpha, \theta) = 1 - \mathcal{F}(\rho, |\phi\rangle). \tag{10}$$

The entangled state experiments had an early termination condition, where the training would stop once the fidelity exceeded 0.999. Fig. 3 shows the average fidelities of the states produced by the derived architectures, and the average NME values after searching, calculated from three repetitions of each experiment. From these results, we see that $\rho$DARTS yields higher fidelity states compared the to the benchmark, qDARTS, which is especially noticeable at higher qubit counts. In these experiments, we kept the total number of quantum simulations fixed for each search type. The fact that $\rho$DARTS reports significantly lower NME values after searching as compared to qDARTS when controlling for the number of quantum simulations, implies that $\rho$DARTS converges faster than qDARTS. Note that the relatively higher NME values reported for $\rho$DARTS with hidden units is due to 21 of the 30 runs terminating early while maintaining relatively high NME values. This implies that with hidden units, $\rho$DARTS can find multiple circuits that produce high fidelity states.

### 5.2 TASK II: VARIATIONAL QUANTUM EIGENSOLVER (VQE)

**Background** The Variational Quantum Eigensolver (VQE) refers to VQAs used to approximate the ground state energies of molecular Hamiltonians (Tilly et al., 2022). VQE is a widely studied problem in quantum computing and is commonly used as a benchmark for QAS algorithms (Wu et al., 2023; Patel et al., 2024; Nakaji et al., 2024; Grimsley et al., 2019).

**Experiment setting** Following Wu et al. (2023), we have employed $\rho$DARTS macro search to approximate the ground state energies of the following molecular Hamiltonians: $H_2$ (4 qubits), LiH (4, 6 qubits) and $H_2O$ (8 qubits); appendix H lists the the molecule configurations for each Hamiltonian. We also benchmark against CRLQAS (Patel et al., 2024), a reinforcement learning approach to QAS, for the same Hamiltonians.

For each simulation, we chose the initial state to be the Hartree Fock state (Slater, 1951) of the molecular Hamiltonian, $\hat{H}$. The loss function of the search is

$$\mathcal{L}(\rho; \alpha, \theta) = \langle\hat{H}\rangle - E_{\text{hf}}, \tag{11}$$

where $\langle\hat{H}\rangle = \text{tr}(\rho\hat{H})$ is the expected energy of $\rho$, and $E_{\text{hf}}$ is the expected energy of the Hartree Fock state. The VQE run is considered successful if the energy error, that is, the difference between $\langle\hat{H}\rangle$ and the true ground state energy, is less than the chemical accuracy required for computational chemistry models to match physical experiments, approximately 0.0016 Hartree (Ha).

Our experiments involved the number of layers being 2, 4, 8 and 16 times the number of qubits, and we observed an exponential relationship between the circuit depth and energy error across our VQE simulations, see appendix H. Table 1 lists the energy error and the circuit depth found by $\rho$DARTS, qDARTS and CRLQAS, where the displayed $\rho$DARTS runs have a comparable depth to the CRLQAS results. Also note that the $\rho$DARTS experiments had an early termination condition, where the training would stop once the energy error fell below $10^{-5}$ Ha.

We continue to see that $\rho$DARTS converges to a solution with fewer quantum simulations as compared to the benchmarks. Patel et al. (2024) report that CRLQAS approaches chemical accuracy for the LiH-4 Hamiltonian after 7,500 training episodes, where each episode involves 1,000 simulations, that is, 7.5 million quantum simulations. Wu et al. (2023) do not specify the number of simulations per epoch, but report that their LiH-4 run had already converged after 3,000 epochs. In contrast, our LiH-4 run in Table 1 reached the early termination condition after 1,095 simulations.

Table 1: Comparing the energy errors and circuit depths of the circuits found for the VQE simulations from our simulations and those reported by Wu et al. (2023) for qDARTS and Patel et al. (2024) for CRLQAS. The energy errors reported are all measured in Hartree (Ha).

| Molecule | $\rho$DARTS (ours) | | qDARTS (Wu et al., 2023) | | CRLQAS (Patel et al., 2024) | |
|---|---|---|---|---|---|---|
| | Energy Error | Depth | Energy Error | Depth | Energy Error | Depth |
| $H_2$-4 | $3.1 \times 10^{-6}$ | 19 | $4.3 \times 10^{-6}$ | 18 | $\mathbf{7.2 \times 10^{-8}}$ | **17** |
| LiH-4 | $9.5 \times 10^{-6}$ | 24 | $1.7 \times 10^{-4}$ | 34 | $\mathbf{2.6 \times 10^{-6}}$ | **22** |
| LiH-6 | $3.0 \times 10^{-4}$ | 44 | $\mathbf{2.9 \times 10^{-4}}$ | 54 | $6.7 \times 10^{-4}$ | **40** |
| $H_2O$-8 | $9.2 \times 10^{-4}$ | 76 | $3.1 \times 10^{-4}$ | **64** | $\mathbf{1.8 \times 10^{-4}}$ | 75 |

## 5.3 TASK III: UNWEIGHTED MAX-CUT

**Background** The unweighted max-cut problem is a combinatorial optimization problem in which the vertices of a graph are partitioned into two subsets with the objective to maximize the number of edges between the different partitions. Although this problem is known to be NP-hard (Karp, 2009), there are many quantum algorithms designed to approximate solutions to the max-cut problem (Farhi et al., 2014; Wang et al., 2018; Amaro et al., 2022). These approaches reformulate max-cut as a Hamiltonian optimization problem. A graph with $n$ vertices is mapped to an $n$-qubit system where the graph's edges, $E$, are used to construct the max-cut Hamiltonian,

$$\hat{H}_c = \sum_{(i,j) \in E} \frac{1}{2} \left( \hat{I} - \hat{Z}^{(i)} \hat{Z}^{(j)} \right). \tag{12}$$

$\hat{H}_c$ is a diagonal matrix, signifying its eigenvectors are the computational basis states $\{ |0 \cdots 00\rangle, \ldots, |1 \cdots 11\rangle \}$ with bit values denoting the partition for which the corresponding vertex is assigned, while the eigenvalues represent the number of edges between the two partitions. The max-cut partition corresponds to the state with the largest eigenvalue. See appendix I to see how the basis states are interpreted as as graph partitions.

**Experiment setting** We used $\rho$DARTS to search for circuits that map the initial state $|\psi_0\rangle = \frac{1}{\sqrt{2^n}} \sum_{k=0}^{2^n-1} |k\rangle$, described as a uniform superposition over all graph partitions, to the max-cut state. In maximizing the expectation value, $\langle \hat{H}_c \rangle = \text{tr}(\rho \hat{H}_c)$, we search for circuits that approximate max-cut solutions. The loss function for this search is

$$\mathcal{L}(\rho; \alpha, \theta) = -\frac{\langle \hat{H}_c \rangle}{|E|}. \tag{13}$$

Our dataset consisted of thirty 10-vertex graphs, each randomly generated using the Erdős–Rényi model (Erdos et al., 1960) with edge creation probabilities of 0.25, 0.5 and 0.75.

In the macro search setting, we searched for circuits containing $m = 15$ layers. For the micro search setting, we defined the super circuit structure to contain subcircuits with $m = 3$ layers acting on $n_s = 2$ qubits. Each edge of a graph corresponds to a subcircuit, as shown in Fig. 4. We also examined our algorithm's robustness to noise, see Fig. 5. In particular, we applied the depolarizing and bit-phase flip noise channels after each of the circuit's layers. Since the qDARTS methodology does not make use of density matrices, it cannot model depolarizing noise, and, therefore, we can only consider bit-phase flip noise in this setting. Furthermore, we are able to assess the quality of the

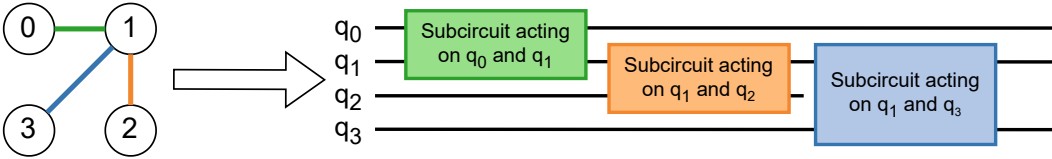

Figure 4: An example of the super circuit structure used in micro search for the max cut problem. The vertices map to the qubits, and edges map to subcircuits acting only on the qubits they connect.

circuits found for each graph by comparing the probability of measuring the max-cut states, $P_m$, and the expectation value normalized by the true max-cut value for each graph, $E_m = \langle \hat{\mathrm{H}}_c \rangle / \max(\hat{\mathrm{H}}_c)$.

Table 2 shows the mean and standard deviation of the metrics $E_m$ and $P_m$ for circuits found in the noiseless simulation for each of the generated graphs. $\rho$DARTS consistently produces states with a higher probability of measuring the true max-cut states in comparison to the baseline, and we found that macro search outperforms micro search.

Table 2: Comparing the max-cut approximations of the circuits produced in the absence of noise.

| Setting | Metric | $\rho$DARTS (ours) | | qDARTS | |
| | | No Hidden Units | Hidden Units | No Hidden Units | Hidden Units |
| --- | --- | --- | --- | --- | --- |
| Macro | $E_m$ | $0.995 \pm 0.020$ | $\mathbf{0.998 \pm 0.011}$ | $0.662 \pm 0.076$ | $0.663 \pm 0.076$ |
| | $P_m$ | $0.933 \pm 0.254$ | $\mathbf{0.966 \pm 0.183}$ | $0.011 \pm 0.019$ | $0.013 \pm 0.023$ |
| Micro | $E_m$ | $\mathbf{0.977 \pm 0.042}$ | $0.960 \pm 0.046$ | $0.668 \pm 0.079$ | $0.662 \pm 0.086$ |
| | $P_m$ | $\mathbf{0.772 \pm 0.381}$ | $0.586 \pm 0.416$ | $0.012 \pm 0.023$ | $0.010 \pm 0.015$ |

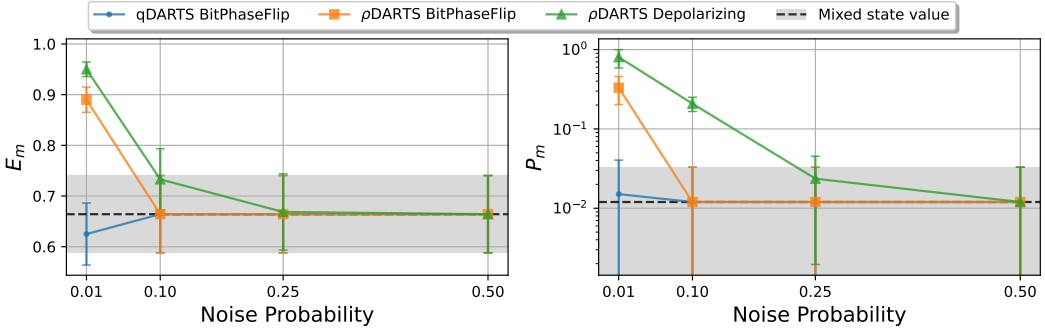

Figure 5: Comparing the max-cut experiments in noisy simulations. The plots show the mean of the metrics $E_m$ and $P_m$, with error bars denoting standard deviation, over macro search experiments with hidden units present. The dashed line and shaded regions denote the mean and standard deviation of the metric values for the maximally mixed state averaged over every graph in the dataset.

## 6 CONCLUSION

We introduced $\rho$DARTS, a differentiable QAS algorithm based on the density matrix formulation of quantum mechanics. Unlike previous approaches that rely on circuit sampling, $\rho$DARTS uses density matrix simulations to enable sampling-free, end-to-end optimization of quantum circuit architectures and parameters. We demonstrated our method to produce PQCs for quantum state initialization, and ground state estimation tasks. We demonstrated that $\rho$DARTS consistently finds optimal architectures, while requiring fewer quantum simulations as compared to the benchmarks, and often outperforms these benchmarks in our experiments.

We also examined $\rho$DARTS under noisy conditions and found that it produced circuits that demonstrated better noise resistance than the benchmark under similar noise settings. Furthermore, because our method makes use of density matrix simulations, it is able to simulate general quantum noise models, like the depolarizing channel, which cannot be simulated with state vectors.

In conclusion, $\rho$DARTS offers new fundamental insights for QAS by aligning DARTS with the mathematical structure of quantum mechanics. The fact that it offers competitive results to alternative QAS methods with significantly fewer quantum simulations demonstrates its potential for further progressing quantum circuit designs for VQAs and quantum machine learning. The limitations of our approach and directions for future work are outlined in appendix A.

REPRODUCIBILITY

To ensure the reproducibility of our work, we have provided a thorough explanation of our method, including the algorithms for the two search settings. We have also provided the details of our qDARTS implementation in appendix C. Experiment settings are provided in the main text for all of our experiments, and the hyperparameter values are listed in appendix E. We provide the compute resources used in appendix F. The source code and datasets for all experiments conducted in this manuscript are provided in the supplementary materials, and we will open source them when the paper is released.

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

## A  LIMITATIONS AND FUTURE WORK

**Scalability**  Since $\rho$DARTS involves simulating the evolution of density matrices, our methodology has increased memory requirements in comparison to state vector simulations; for $n$ qubits, the memory required to store a density matrix scales as $4^n$, but state vectors scale as $2^n$. The fundamentally exponential scaling of quantum simulation sets a practical limit on the number of qubits which can be simulated with our method, and indeed any QAS method reliant on quantum simulation. This limitation warrants further exploration into space efficient simulation strategies for mixed states.

**Reusing circuits**  In this work, we use QAS to generate circuits that are tailored to produce solutions for a single instance of a given problem, for example, in our max-cut experiments, we produce a unique circuit for every graph in our dataset. Although we adopted this paradigm from the methods we benchmarked against, it severely limits the applicability of QAS. Future work will focus on finding circuit architectures that can be reused for multiple instances of the same problem. A first step towards this approach would involve using micro search to find reusable subcircuits that can be trained to find the max-cut partitions of multiple graphs, with varying qubit counts, akin to gadget reinforcement learning approaches to QAS (Kundu & Sarra, 2024; Olle et al., 2025).

**Validation on quantum hardware**  In this work, we do not validate the circuits found by our QAS methods on real quantum computers. Future work will include validation on real hardware, with focus on simulating realistic noise models and hardware-native gate sets.

## B  QUANTUM COMPUTING FUNDAMENTALS

### B.1  QUANTUM STATES

The state of a quantum bit, qubit, is a 2-dimensional unit complex vector:

$$|\psi\rangle = \begin{bmatrix} \psi_0 \\ \psi_1 \end{bmatrix} = \psi_0 |0\rangle + \psi_1 |1\rangle,$$

$$\psi_0, \psi_1 \in \mathbb{C}, \quad |\psi_0|^2 + |\psi_1|^2 = 1. \tag{14}$$

The quantities $|\psi_0|^2$ and $|\psi_1|^2$ denote the probabilities of measuring the qubit as 0 and 1, respectively. Similarly, the state of $n$ qubits is an $N = 2^n$-dimensional unit complex vector

$$|\psi\rangle = \begin{bmatrix} \psi_0 \\ \psi_1 \\ \vdots \\ \psi_{N-1} \end{bmatrix} = \sum_{k=0}^{N-1} \psi_k |k\rangle, \tag{15}$$

where the quantity $|\psi_k|^2$ denotes the probability of measuring the the qubit to be the $n$-bit representation of the integer $k$.

### B.2  QUANTUM GATES

Quantum states evolve with the application of quantum gates. These quantum gates are defined as unitary matrices $\hat{U}$ such that the evolved state is $|\psi_{t+1}\rangle = \hat{U} |\psi_t\rangle$. Common quantum gates include the well-known Pauli gates:

$$\hat{X} = \begin{bmatrix} 0 & 1 \\ 1 & 0 \end{bmatrix}, \; \hat{Y} = \begin{bmatrix} 0 & -i \\ i & 0 \end{bmatrix}, \; \hat{Z} = \begin{bmatrix} 1 & 0 \\ 0 & -1 \end{bmatrix}. \tag{16}$$

Quantum gates can also have trainable parameters, for example, the Pauli rotation gates:

$$\hat{R}_x(\theta) = \begin{bmatrix} \cos\frac{\theta}{2} & -i\sin\frac{\theta}{2} \\ -i\sin\frac{\theta}{2} & \cos\frac{\theta}{2} \end{bmatrix}, \; \hat{R}_y(\theta) = \begin{bmatrix} \cos\frac{\theta}{2} & -\sin\frac{\theta}{2} \\ \sin\frac{\theta}{2} & \cos\frac{\theta}{2} \end{bmatrix}, \; \hat{R}_z(\theta) = \begin{bmatrix} e^{-i\theta/2} & 0 \\ 0 & e^{i\theta/2} \end{bmatrix}. \tag{17}$$

These matrices acan be extended to multi-qubit systems using the tensor product. For example, a two-qubit gate consisting of $\hat{Z}$ acting on qubit 0 and $\hat{X}$ acting on qubit 1 is given by

$$\hat{X}^{(1)} \otimes \hat{Z}^{(0)} = \begin{bmatrix} 0 & 0 & 1 & 0 \\ 0 & 0 & 0 & -1 \\ 1 & 0 & 0 & 0 \\ 0 & -1 & 0 & 0 \end{bmatrix}. \tag{18}$$

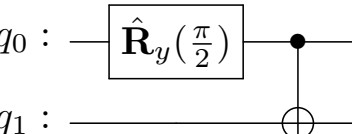

Figure 6: Quantum circuit that maps the input state $|00\rangle$ to the Bell state $|\Phi\rangle = \frac{|00\rangle + |11\rangle}{\sqrt{2}}$. The second gate is a CNOT gate with control $c = 0$ and target $t = 1$.

Another common two-qubit gate is the Controlled-Not (CNOT) gate. This gate acts on two qubits, the control qubit $c$ and the target qubit $t$ such that the $\hat{X}$ gate is applied to the target if the control qubit is in state $|1\rangle$. For example, the CNOT gate acting on a two qubit system where $c = 1$ and $t = 0$ maps the basis states as follows:

$$
\begin{aligned}
\hat{\text{CNOT}}_{0,1} |00\rangle &= |00\rangle\,, \\
\hat{\text{CNOT}}_{0,1} |01\rangle &= |01\rangle\,, \\
\hat{\text{CNOT}}_{0,1} |10\rangle &= |11\rangle\,, \\
\hat{\text{CNOT}}_{0,1} |11\rangle &= |10\rangle\,.
\end{aligned}
\tag{19}
$$

A sequence of quantum gates is called a quantum circuit. For instance, the quantum circuit that maps the initial state $|00\rangle$ to the Bell state $|\Phi\rangle = \frac{1}{\sqrt{2}} |00\rangle + \frac{1}{\sqrt{2}} |11\rangle$ is shown in Fig. 6.

### B.3 VARIATIONAL QUANTUM ALGORITHMS

Variational Quantum Algorithms (VQAs) are a family of hybrid-quantum algorithms that involve optimizing the parameters of a fixed Parametric Quantum Circuit (PQC). The PQC is applied to a quantum computer, and the measurement statistics are used to calculate the value of the loss function and the gradients, with respect to each of the gate parameters. These gradients are then passed to a classical optimizer, which updates the gate parameters. This process is repeated until the loss converges.

## C QUANTUMDARTS SPECIFICATION

To demonstrate the effectiveness of our algorithm, we used qDARTS (Wu et al., 2023) as a benchmark in our experiments. We implemented the algorithm according to the authors' specifications since the source code was not made available. We now present the specifics of our qDARTS implementation.

### C.1 ALGORITHM SUMMARY

The architecture parameters $\alpha \in \mathbb{R}^{m \times n \times |\mathcal{G}|}$ give rise to softmax probability distributions, $\mathcal{P}_{ij}$, for selecting gates for each position in the circuit from the candidate gate set $\mathcal{G}$. Using the Gumbel-softmax reparameterization trick (Jang et al., 2016), a single gate $\hat{U}_{ij}$ is sampled for each position as follows:

$$
\hat{U}_{ij} = \sum_{k=0}^{|\mathcal{G}|-1} h_{ij}^{(k)} \mathcal{G}_k,
\tag{20}
$$

$$
\text{where } h_{ij} = \text{one-hot}(\arg\max_k (\alpha_{ij}^{(k)} + G_k)).
$$

Each $G_k$ are independent random variables all sampled from the Gumbel distribution $G = -\ln(-\ln(X))$, where $X \sim \mathcal{U}(0,1)$. $h_{ij}$ is used in the forward pass to calculate the loss values, but because argmax is not differentiable, it is replaced with the following soft sampling

---

**Algorithm 3** qDARTS macro search

---

**Require:** number of qubits $n$, number of layers $m$, candidate gate set $\mathcal{G}$, Randomly initialized $\alpha$ and $\theta$, initial state $|\psi_0\rangle$, $num\_epochs$, $\tau$, $num\_iter$
    **for** $epoch \leftarrow 1$ to $num\_epochs$ **do**
        $|\psi\rangle \leftarrow |\psi_0\rangle$
        **for** $i \leftarrow 0$ to $m - 1$ **do**
            **for** $j \leftarrow 0$ to $n - 1$ **do**
                Obtain $\hat{U}_{ij}$ from the Gumbel-softmax trick
                $|\psi\rangle \leftarrow \hat{U}_{ij} |\psi\rangle$
            **end for**
        **end for**
        **for** $iter \leftarrow 1$ to $num\_iter$ **do**
            Calculate loss $\mathcal{L}_\theta(|\psi\rangle\,;\theta)$
            Update $\theta$ by gradient descent
        **end for**
        Calculate loss $\mathcal{L}_\alpha(|\psi\rangle\,;\alpha)$
        Update $\alpha$ by gradient descent
    **end for**
    Fix the final circuit architecture $\mathcal{A}^* \in \mathcal{S}$ such that $\mathcal{A}_{ij}^* = \arg\max_k \mathcal{P}_{ij}^{(k)}$

---

expression in the backward pass:

$$\tilde{h}_{ij}^{(k)} = \frac{\exp\left((\ln(\mathcal{P}_{ij}^{(k)}) + G_k)/\tau\right)}{\sum_{k'=0}^{|\mathcal{G}|-1} \exp\left((\ln(\mathcal{P}_{ij}^{(k')}) + G_{k'})/\tau\right)}, \tag{21}$$

where the temperature $\tau$ is a hyperparameter. In the limit $\tau \to 0$, the soft sampling is equivalent to hard sampling. Once a circuit is sampled, the gate parameters are updated using gradient descent for a fixed number of iterations, $num\_iter$, before finally updating the architecture parameters. This process is summarized in Algorithm 3. If the number of quantum simulations is fixed to be $N_q$, then qDARTS runs for $N_q/num\_iter$ epochs.

### C.2 IMPLEMENTATION DETAILS

We implemented qDARTS in PyTorch (Imambi et al., 2021), which has an in-built method, `torch.nn.functional.gumbel_softmax`, for the Gumbel-softmax trick that automatically handles the hard and soft sampling in the forward and backward passes.

Note that qDARTS requires two loss functions, $\mathcal{L}_\alpha$ for the architecture parameters and $\mathcal{L}_\theta$ for the gate parameters. In our experiments, $\mathcal{L}_\alpha$ included the VQA loss function with the entropy regularization term, and $\mathcal{L}_\theta$ included the VQA loss function with the angle penalty term.

## D COMPLEXITY

The forward pass of the $\rho$DARTS algorithm involves repeatedly applying quantum operations, $\mathcal{E}$, which are defined in Eq. 4 and Eq. 6 for the macro and micro search settings, respectively. Each operation $\mathcal{E}(\rho)$ is computed by taking a weighted sum of the outputs of applying every gate in $\mathcal{G}$ to the state $\rho$. Our implementation of this operation makes use of a custom GPU kernel that computes the outputs of each gate in parallel, where each thread computes one element of the resultant density matrices. To simulate $n$ qubits, with the assumption that the GPU can process $M$ threads in parallel, the time complexity of computing $\mathcal{E}$ is

$$\mathrm{T}_{\mathcal{E}}(n, |\mathcal{G}|) = O\left(\frac{|\mathcal{G}|4^n}{M}\right). \tag{22}$$

It follows that the time complexity of macro search with $n$ qubits, $m$ layers, and $T$ epochs, noting that $|\mathcal{G}| = n + 3$ in our gate set, is

$$\text{T}_{\text{macro}} = O(Tmn\, \text{T}_{\mathcal{E}}(n, n+3)) = O\left(\frac{Tmn^2 4^n}{M}\right). \tag{23}$$

Similarly, the time complexity of micro search with $n$ qubits, $N_c$ subcircuits each with $m$ layers and acting on $n_s$ qubits, which implies $|\mathcal{G}| = n_s + 3$, and $T$ epochs is

$$\text{T}_{\text{micro}} = O(TN_c mn_s\, \text{T}_{\mathcal{E}}(n, n_s+3)) = O\left(\frac{TN_c mn_s^2 4^n}{M}\right). \tag{24}$$

Computing the operations $\mathcal{E}$ involves allocating space for the density matrix outputs of each gate, and for the weighted sum of these outputs. This implies that each $\mathcal{E}$ allocates additional memory of the size

$$\text{SPACE}_{\mathcal{E}}(n, |\mathcal{G}|) = O((|G|+1)4^n) = O(|\mathcal{G}|4^n). \tag{25}$$

It follows that the space complexity of macro search is

$$\text{SPACE}_{\text{macro}} = O\left(mn\, \text{SPACE}_{\mathcal{E}}(n, n+3)\right) = O\left(mn^2 4^n\right), \tag{26}$$

and the space complexity of micro search is

$$\text{SPACE}_{\text{micro}} = O\left(N_c mn_s\, \text{SPACE}_{\mathcal{E}}(n, n_s+3)\right) = O\left(N_c mn_s^2 4^n\right). \tag{27}$$

## E  HYPERPARAMETERS

Table 3 lists the hyperparameter values chosen for our experiments. The parameter $T_{max}$ of the cosine annealing learning rate scheduler is calculated from $T_f$ in the table as $T_{max} = T_f \times num\_epochs$.

Table 3: Hyperparameter values used in our experiments.

| Parameter | Entangled State | Image State (6 qubit) | Image State (8 qubit) | VQE | Max-cut |
|---|---|---|---|---|---|
| $num\_epochs$ | 10,000 | 1,000 | 10,000 | $[1,000, 5,000, 10,000]$ | 1,000 |
| learning rate | 0.01 | 0.1 | 0.1 | $[0.1, 0.01, 0.001]$ | 0.1 |
| $T_f$ | 0.1 | 0.1 | 0.1 | $[0.1, 0.5, 0.75, 1.0]$ | 0.1 |
| $s_0$ | 0.0 | 0.0 | 0.0 | 0.0 | 0.0 |
| $s_1$ | 0.1 | 0.1 | 0.1 | 0.1 | 0.1 |
| $s_\theta$ | 0.01 | 0.01 | 0.01 | 0.01 | 0.01 |
| hidden units? | [✗, ✓] | ✓ | ✓ | ✓ | [✗, ✓] |
| qDARTS Specific Parameters | | | | | |
| $N_q$ | 10,000 | 1,000 | 10,000 | - | 10,000 |
| $num\_iter$ | [1, 10] | 10 | 10 | - | 10 |
| $\tau$ | 0.05 | 0.05 | 0.05 | - | 0.05 |

## F  COMPUTE RESOURCES

Table 4 lists the compute resources used in our QAS experiments.

## G  SUPPLEMENTS TO THE STATE INITIALIZATION TASK

### G.1  QUANTUM CIRCUIT FOR GHZ STATES

The quantum circuit to generate the 3-qubit GHZ state is shown in Fig. 7. This can be generalized to $n > 3$ qubits by adding additional CNOT gates targeting the additional qubits, like how the circuit in Fig. 6 generalizes to the circuit in Fig. 7.

Table 4: Compute resources used for our experiments

| Experiments | CPUs used | GPUs used |
| --- | --- | --- |
| Entangled state initialization, noise-free max-cut | 8x Intel Xeon @ 2.20 GHz | 1x NVIDIA A100 40GB SXM |
| Image state initialization, VQE, noisy max-cut | 12x Intel Xeon Platinum 8568Y+ @ 2.30 GHz | 1x NVIDIA H200 141GB |

In our circuit search space, each layer consists of a sequence of gates ordered by the qubit index that they target. Under this structure, the GHZ circuits can fit in a circuit with exactly one layer. However, the circuits that can generate the GHZ state in one layer form a very small part of the search space. The first gate must be $\hat{R}_y$, the next gate must be a CNOT whose control is the first qubit. The next gate must also be a CNOT gate whose control is any of the previous two qubits, and so on. The fraction of circuits that can produce GHZ states in the search space is

$$\mathcal{P}_{GHZ} = \frac{(n-1)!}{(n+3)^n},\tag{28}$$

which decays very quickly as the number of qubits increases.

Further, with the initial state $|\psi_0\rangle = |0\rangle^{\otimes n}$, every gate other than $\hat{R}_x$ and $\hat{R}_y$ acts as the identity operator. This means that a circuit without any $\hat{R}_x$ and $\hat{R}_y$ gates is equivalent to the identity, up to a global phase, and the fraction of such circuits is

$$\mathcal{P}_{id} = \left(\frac{n+1}{n+3}\right)^n.\tag{29}$$

As such, the identity circuits become a local minima during the search, with the fidelity $\mathcal{F}(|\psi_0\rangle, |GHZ\rangle) = 0.5$. Increasing the number of layers in the search allows for a larger fraction of the search space to generate the GHZ state. Fig. 8 shows that adding layers decreases the probability of $\rho$DARTS to select a circuit equivalent to the identity. Figs. 14 and 16 show the circuits found by $\rho$DARTS that generate the GHZ states.

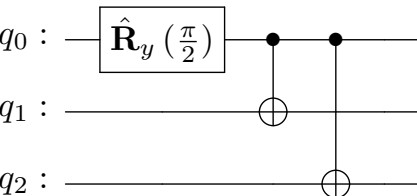

Figure 7: Quantum circuit that maps the input state $|000\rangle$ to the GHZ state.

## G.2 QUANTUM CIRCUIT FOR W STATES

The quantum circuits to generate the 3- and 4-qubit W states are shown in Figs. 9a and 9b. These are generalized to higher qubit counts by adding additional controlled-$\hat{R}_y$ and CNOT gates. The controlled-$\hat{R}_y$ gates can be decomposed into the gates in our chosen gate set as shown in Fig. 10a. Figs. 10b and 10c show the 3- and 4-qubit W state circuits decomposed into our gate set, and the pattern is established that the $n$-qubit W state circuit can fit in a circuit with $3(n-1)$ layers. Figs. 15, 17a, 17b, 18 and 19 show the circuits found by $\rho$DARTS to generate the W states.

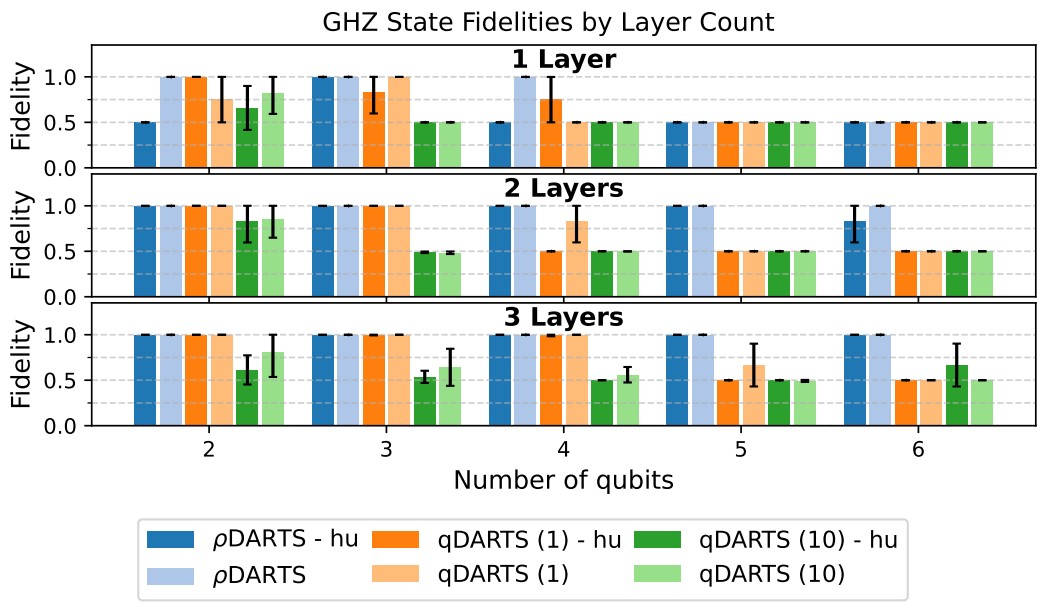

Figure 8: The fidelities of the states found by $\rho$DARTS, and qDARTS when initializing GHZ states for different layers counts, averaged over three runs. Error bars indicate one standard deviation from the mean, clipped between 0 and 1.

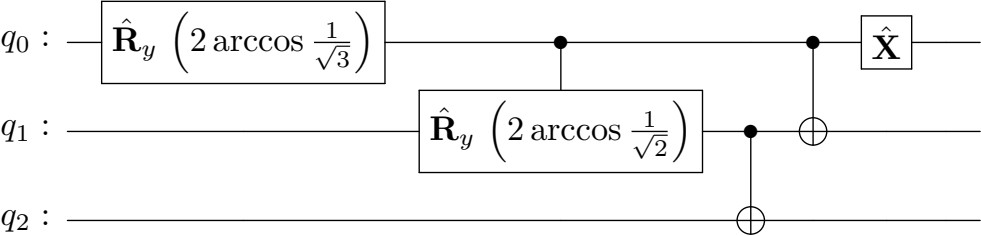

(a) Quantum circuit that maps the input state $|000\rangle$ to the 3-qubit W state.

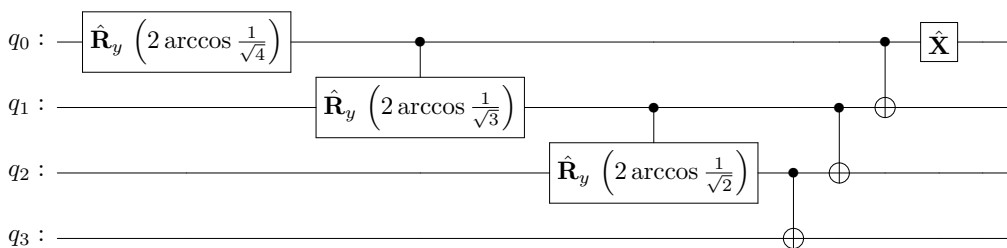

(b) Quantum circuit that maps the input state $|0000\rangle$ to the 4-qubit W state.

Figure 9: Quantum circuits that generate the 3 and 4-qubit W states.

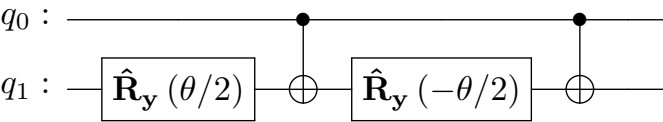

(a) Decomposition of the controlled-$\hat{R}_y$ gate using $\hat{R}_y$ and CNOT gates.

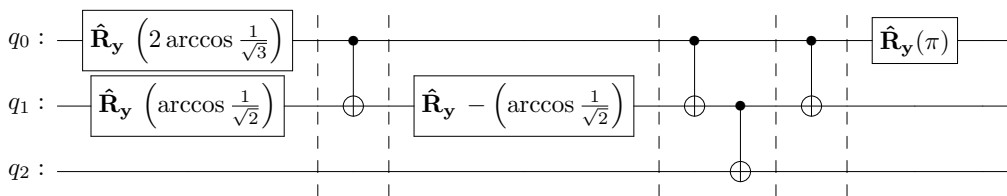

(b) Quantum circuit that maps the input state $|000\rangle$ to the 3-qubit W state, fits in 6 layers.

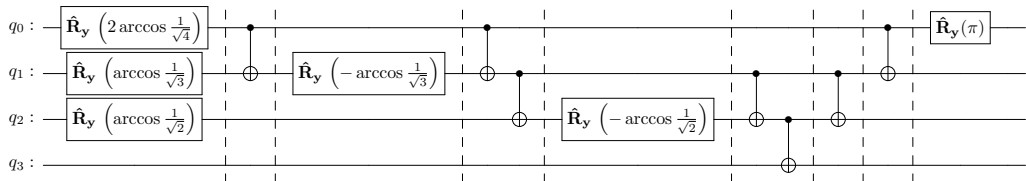

(c) Quantum circuit that maps the input state $|0000\rangle$ to the 4-qubit W state, fits in 9 layers.

Figure 10: Quantum circuits that generate W states decomposed into the gates in $\mathcal{G}$.

### G.3 AMPLITUDE ENCODED IMAGES

Amplitude encoding refers to encoding a data vector $\vec{x} \in \mathbb{R}^N$ into the quantum state vector of $\lceil \log_2 N \rceil$ qubits. The amplitude encoded state is

$$|\psi(\vec{x})\rangle = \sum_{j=0}^{N-1} \frac{x_j}{\|\vec{x}\|_2} |j\rangle. \tag{30}$$

To compute the amplitude encode state of an $N \times N$ pixel grayscale image, $I$, we first normalize the pixels values between 0 and 1 and unwrap the pixel values into a vector $\vec{x}(I) \in \mathbb{R}^{N^2}$. Then we encode this vector according to Eq. 30 in an $n = \lceil 2 \log_2 N \rceil$-qubit state vector, $|\psi(I)\rangle$.

Given an $n$-qubit state vector $|\psi\rangle$ used to approximate the image $I$, we reconstruct the approximated image using the measurement probabilities of $|\psi\rangle$. Let $\vec{p}$ be the vector of measurement probabilities, $p_i = |\psi_i|^2$, then its square root gives us a unit vector $\vec{x}(|\psi\rangle) = \sqrt{\vec{p}} \in \mathbb{R}^{2^n}$. Multiplying this unit vector by the image norm $\|\vec{x}(I)\|_2$ gives us the normalized pixel values of the approximated image.

Fig. 13 shows the reconstructed MNIST images found in our image state experiments.

## H SUPPLEMENTS TO THE VQE TASK

### H.1 MOLECULAR HAMILTONIANS

Table 5 lists the configurations of each molecule simulated along with the transformation used to encode the fermionic Hamiltonians to qubit Hamiltonians. The OpenFermion (McClean et al., 2020) Python library can be used to generate the molecular Hamiltonians with this information.

Table 5: List of molecules considered in our simulations, following directly from Wu et al. (2023).

| Molecule | # qubits | Fermion to qubit mapping | Nuclear Coordinates (Å) |
|---|---|---|---|
| $H_2$ | 4 | Jordan-Wigner (Jordan & Wigner, 1928) | H - (0, 0, -0.35)
H - (0, 0, 0.35) |
| LiH | 4 | Parity (Seeley et al., 2012) | Li - (0, 0, 0)
H - (0, 0, 2.2) |
| LiH | 6 | Jordan-Wigner | Li - (0, 0, 0)
H - (0, 0, 2.2) |
| $H_2O$ | 8 | Jordan-Wigner | H - (-0.021, -0.002, 0)
O - (0.835, 0.452, 0)
H - (1.477, -0.273, 0) |

### H.2 RELATING THE DEPTH AND ENERGY ERROR OF THE FOUND CIRCUITS

We ran multiple $\rho$DARTS searches with different layer counts for each molecule in the VQE experiments. We have plotted the circuit depth and energy errors for the runs that reached an energy error below the chemical accuracy in Fig. 11, and observe an exponential relationship between these metrics. Note that the clustering of points around the energy error $10^{-5}$ with different circuit depths is due to the early termination condition of our search. We believe that without the early termination, these runs would report lower energy errors, with smaller errors for deeper circuits.

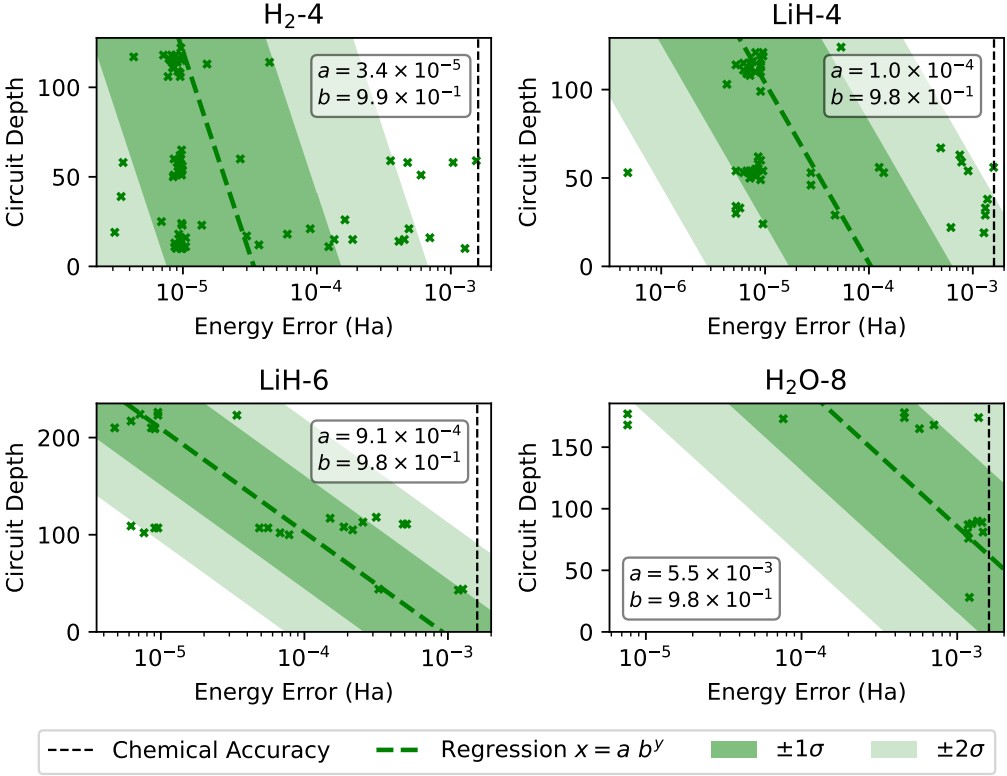

Figure 11: Plots showing the energy errors (x-axis) and depths (y-axis) found by $\rho$DARTS for VQE simulations. The green dashed line is an exponential regression between the energy error and circuit depth with the shaded regions surrounding it indicating one and two standard deviations of the residuals respectively. The chemical accuracy is also marked with the vertical dashed line.

# I SUPPLEMENTS TO THE MAX-CUT TASK

## I.1 INTERPRETING STATES AS GRAPH PARTITIONS

In the quantum formulation of the max-cut problem, the basis states $\{|0\cdots00\rangle,\ldots,|1\cdots11\rangle\}$ correspond to the ways to the ways to partition a the vertices of a graph. Fig. 12 shows a few examples of bit strings and their corresponding graph partitions. The quantum circuits found by $\rho$DARTS can be evaluated according to the measurement distributions of the states they produce.

The metric $P_m$ denotes the probability of measuring the produced state to be any of the max-cut partitions. This provides us with insight as to how often the circuits we generate hone in on the exact solutions. On the other hand, the metric $E_m = \langle\hat{H}_c\rangle / \max(\hat{H}_c)$ denotes the expected number of edges between the partitions in all of the measurement outcomes, normalized by the number of edges in the max-cut partition. This provides a measure for how close our max-cut approximations are to the true solutions.

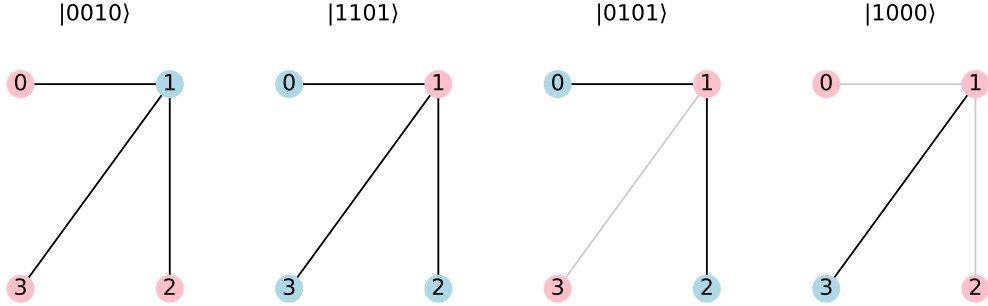

Figure 12: A few examples showing bit strings and their corresponding graph partitions. Vertices corresponding to the bit value 1 are colored light blue and those corresponding the bit value 0 are pink. Edges between the two partitions are colored black and the edges within the same partition are colored light gray.

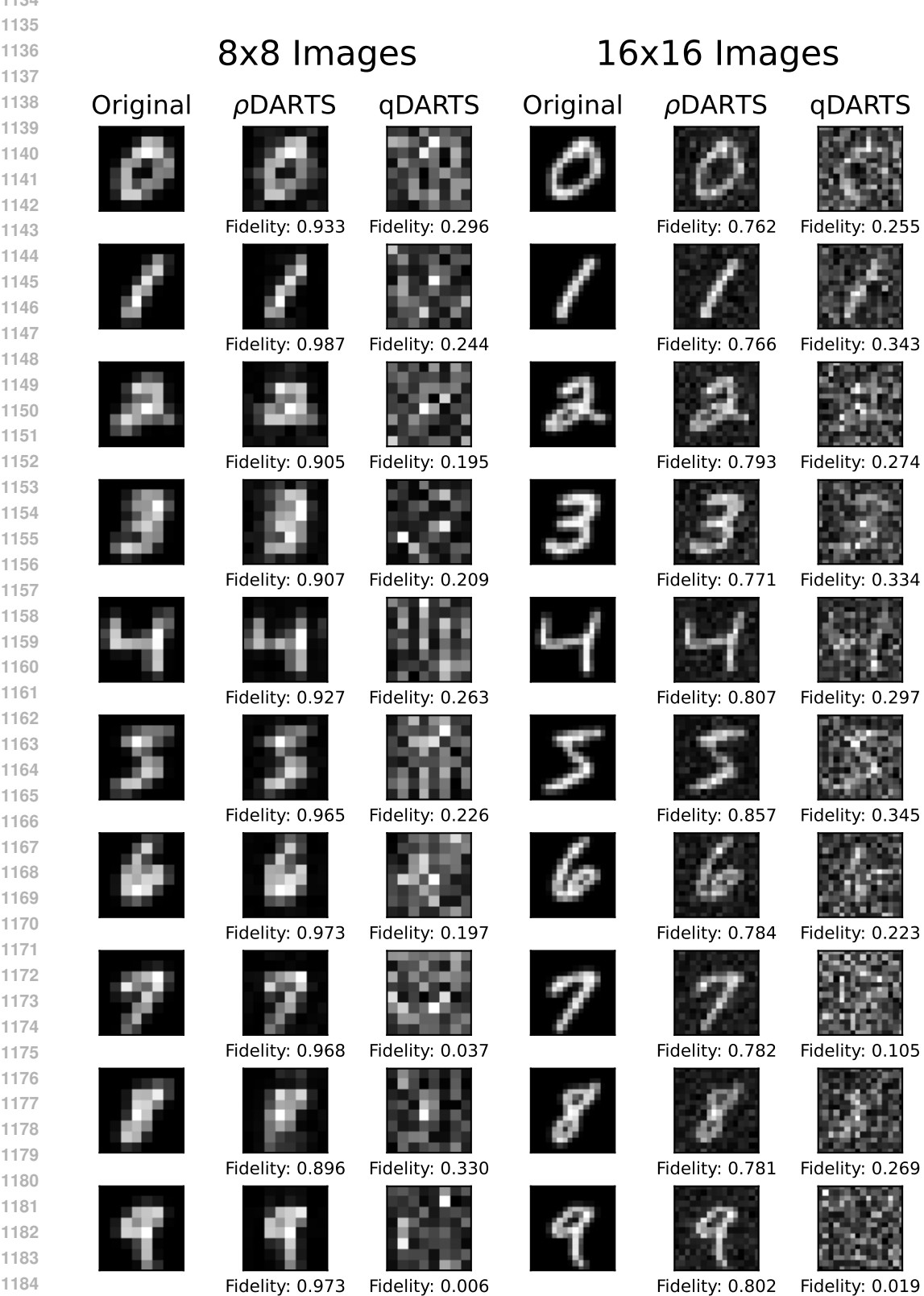

Figure 13: MNIST image reconstructions found by $\rho$DARTS and qDARTS.

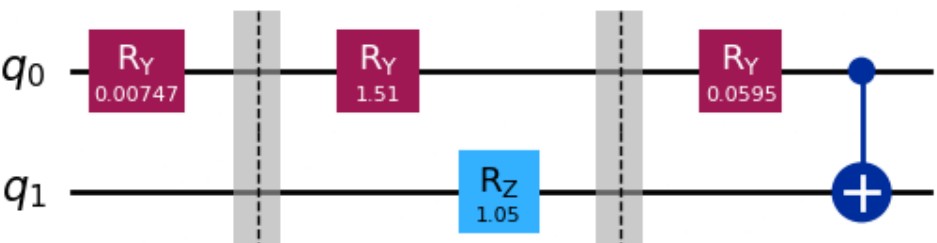

(a) Circuits found by $\rho$DARTS that prepare the 2-qubit GHZ state, i.e. the Bell state $|\Phi\rangle$.

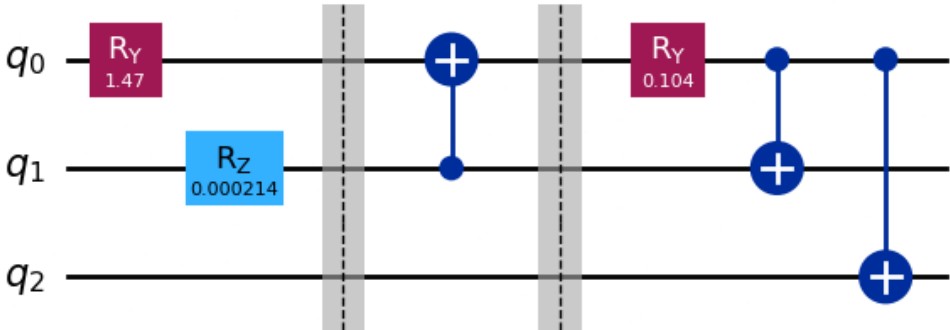

(b) Circuits found by $\rho$DARTS that prepare the 3-qubit GHZ state.

Figure 14: Circuits found by $\rho$DARTS that prepare the 2 and 3-qubit GHZ states.

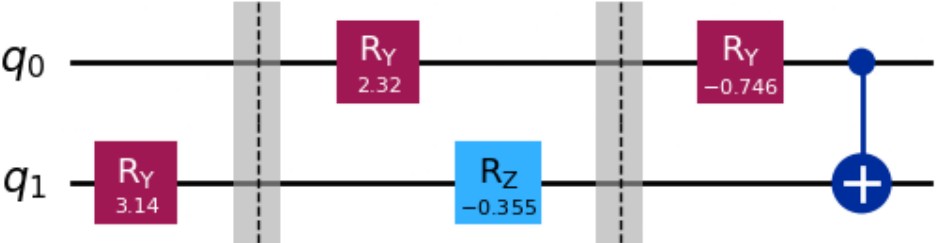

Figure 15: Circuit found by $\rho$DARTS that prepares the 2-qubit W state, i.e. the Bell state $|\Psi\rangle$

1242
1243
1244
1245
1246
1247
1248
1249
1250
1251
1252
1253
1254
1255
1256
1257
1258
1259
1260
1261
1262
1263
1264
1265
1266
1267
1268
1269
1270
1271
1272
1273
1274
1275
1276
1277
1278
1279
1280
1281
1282
1283
1284

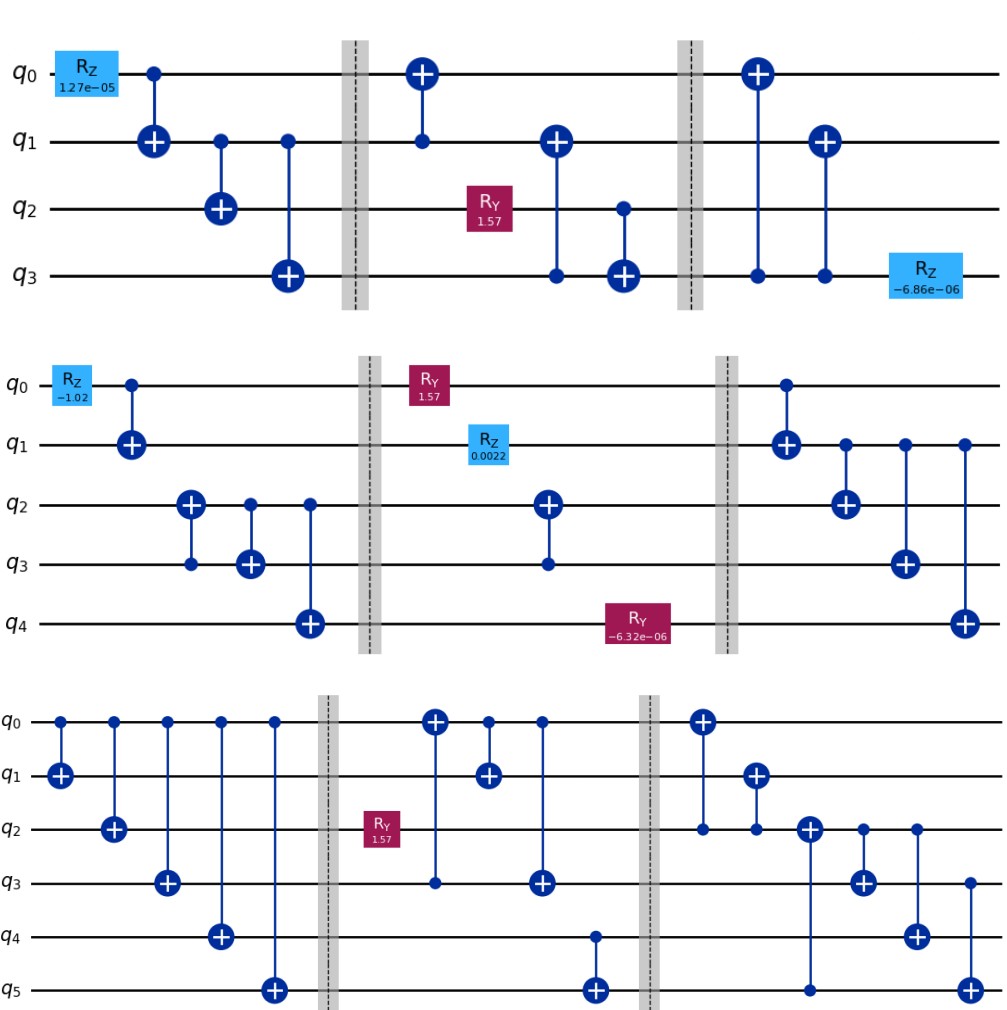

Figure 16: Circuits found by $\rho$DARTS that prepare the 4, 5 and 6-qubit GHZ states.

1285
1286
1287
1288
1289
1290
1291
1292
1293
1294
1295

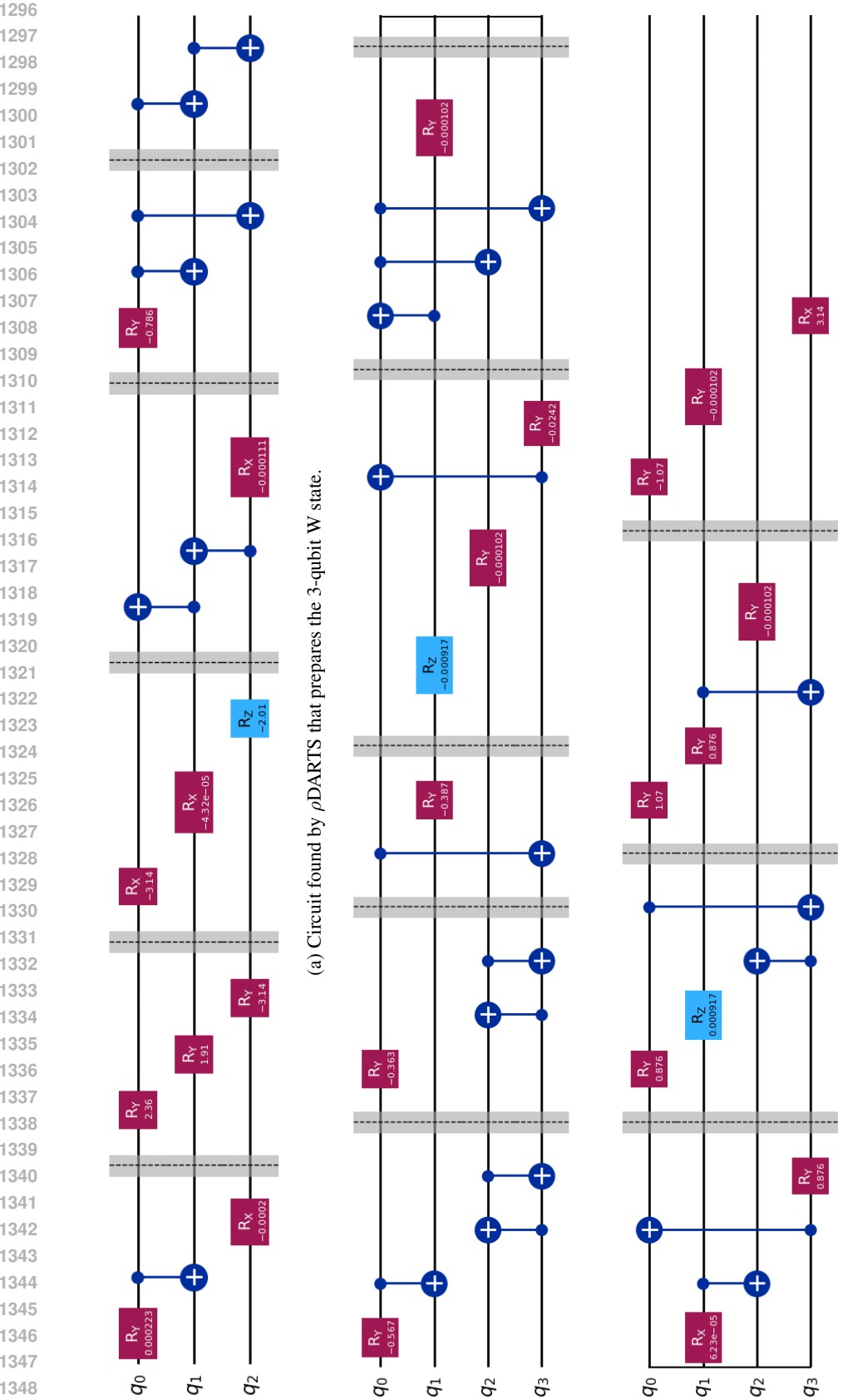

(a) Circuit found by $\rho$DARTS that prepares the 3-qubit W state.

(b) Circuit found by $\rho$DARTS that prepares the 4-qubit W state.

Figure 17: Circuits found by $\rho$DARTS that prepare the 3 and 4-qubit W states.

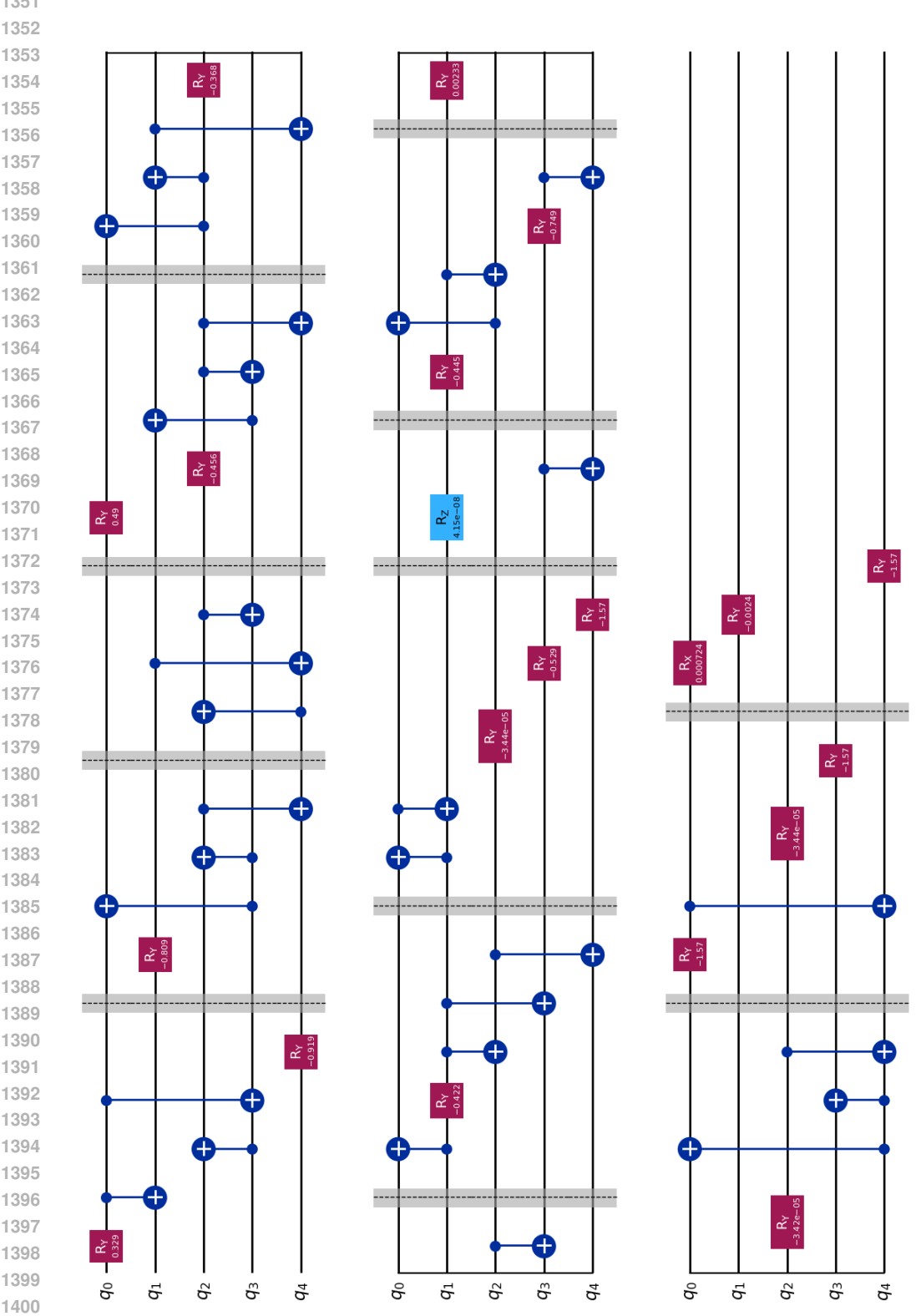

Figure 18: Circuit found by $\rho$DARTS that prepares the 5-qubit W state.

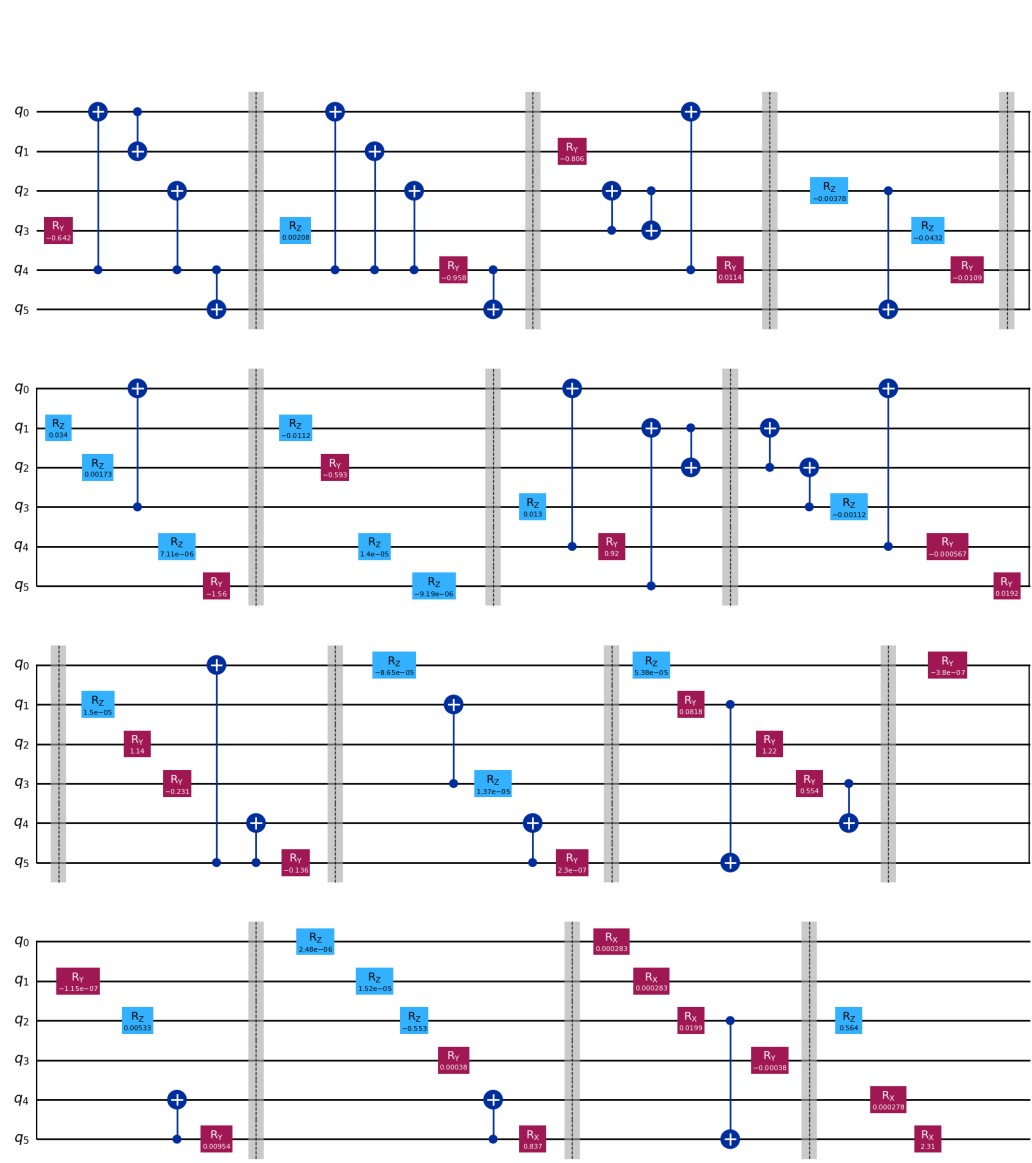

Figure 19: Circuit found by $\rho$DARTS that prepares the 6-qubit W state.

