# OpenReview forum: "Simulating Mixed State Dynamics to Enable Differentiable Quantum Architecture Search"
_ICLR.cc/2026/Conference — Submitted to ICLR 2026_

### Official Review · Reviewer_51Km · 2025-10-31

**Soundness:** 3
**Presentation:** 3
**Contribution:** 2
**Rating:** 4
**Confidence:** 4

**Summary:**

This paper introduces $\rho$DARTS, a differentiable quantum architecture search (QAS) algorithm based on the density matrix formalism. The authors' motivation stems from a theoretical observation: the original DARTS "relaxation" (a probabilistic sum of operations) is physically invalid for quantum state vectors. While prior work (e.g., qDARTS) addressed this by using Gumbel-Softmax to enable differentiable sampling, this paper proposes an alternative. It returns to the original DARTS "summation" idea by using density matrices, where a probabilistic sum (convex combination) is physically valid. This allows $\rho$DARTS to create a single mixed state representing the entire search ensemble, enabling a "sampling-free" optimization. A key benefit of this formalism is the natural ability to incorporate arbitrary noise models, such as the depolarizing channel. The authors present experiments on state initialization, VQE, and Max-Cut to demonstrate their method's efficacy.

**Strengths:**

1. Theoretically Sound Formulation: The core premise of the paper is well-founded. It correctly identifies that the density matrix formalism is the quantum-mechanically proper way to execute the original DARTS-style "relaxation" (i.e., a weighted summation over all possible operations), which is not possible with state vectors. This "sampling-free" approach is a clean and logical alternative to the sampling-based path taken by methods like qDARTS.

2. Native Noise-Aware Search Capability: A significant and practical advantage of the density matrix approach is the ability to naturally incorporate arbitrary quantum noise channels (as defined in Eq. 1). This is a clear benefit over state-vector-based methods, which cannot easily model incoherent, non-unitary noise like the depolarizing channel. The noise-robustness experiments (Fig. 5) are a strong point of the paper.

**Weaknesses:**

1. The Discretization Gap: The method optimizes the parameters $(\alpha, \theta)$ of a mixed state $\rho'$, which represents a probabilistic ensemble of all circuits. However, to obtain the final architecture, it applies a "hard" argmax operation (Algorithm 1, line 14) after training. This creates a "discretization gap" common to DARTS-like methods. There is no guarantee that the performance of the final, discrete circuit $\mathcal{A}^*$ will match the optimized performance of the mixed state $\rho'$.
2. Highly Incremental Contribution: The paper's methodological novelty is limited. It heavily borrows its entire experimental framework, including the "macro search" and "micro search" concepts, directly from prior work (qDARTS, Wu et al., 2023). The only substantial change is the replacement of the Gumbel-Softmax sampling component with the density matrix summation component. While this change is logical, it makes the paper feel highly derivative and more like an incremental follow-up rather than a novel contribution.
3. Failure to Address the Core QAS Challenge: The quantum computing field has evolved significantly since 2023. The core, unresolved challenge for all simulation-based QAS is the $O(2^n)$ exponential scaling wall, which restricts them to classically-trivial problems. The QAS field now needs fundamental breakthroughs that solve this scaling problem. This paper, while theoretically neat, fails to provide this. In fact, it regresses on the most critical axis by introducing an $O(4^n)$ computational cost, making it less scalable than its predecessors. This work does not fundamentally change the game for QAS, making it less attractive for the general ML community.

**Questions:**

No questions.

---

> ### Author Response · Authors · 2025-11-20
>
> The reviewer is correct to point out that the final hard-sampling step can introduce a discretization gap which is common to DARTS-like methods. In our method such a gap would correspond to the probabilistic ensemble of circuits defined by the parameters $(\alpha, \theta)$ giving rise to a mixed state $\rho$ that satisfies the optimization problem, but the pure state obtained by running the sampled circuit from the ensemble does not satisfy the problem. We address this point in two parts:
> 1. In our experiments, the objective function, $f$, for the optimization problems was either the fidelity between states or the expectation value of an operator. The search process generates the mixed state $\rho = \sum_ \mathcal A p_\mathcal A \ket{\psi_\mathcal{A}}\bra{\psi_\mathcal{A}}$, and it is the case that $$f(\rho) = \sum_\mathcal A p_\mathcal A f(\ket{\psi_\mathcal A}) =  \mathbb{E}\_\mathcal A[f(\ket{\psi_\mathcal A})].$$
> This implies that the distribution of $f$ values we can observe is centered around $f(\rho)$. Moreover, since we use argmax to sample the most-likely circuit from the search space, the $f$ value obtained should be close to $f(\rho)$.
> 2. We introduced a regularization term based on the entropies of the gate distributions defining the ensemble of circuits specifically to address the discretization gap in lines 246-266. The NME regularization guides the distributions towards zero-entropy distributions by the end of the search. By adding this term to the overall loss function, we promote the search process to collapse the probabilistic ensemble of architectures to a single architecture $\mathcal A^*$, thereby eliminating the discretization gap entirely. By tracking the NME value over time, we can get a good idea of when this collapse occurs, and we choose hyperparameters that force this collapse during the search.
>
> The reviewer has claimed that the novelty of our work is limited because our experimental framework matches that of qDARTS. The fact is that qDARTS and RhoDARTS are two different approaches to port DARTS to a quantum setting. Because they are both DARTS-like algorithms, it is natural to directly compare the two approaches through experiments, and by doing so we have shown our method to outperform qDARTS. The shared experimental framework is a consequence of comparing similar algorithms, however the novelty and contribution of our work is the method itself.
>
> Although the reviewer recognizes that the novelty of our work is the sampling-free approach to QAS, they seem to trivialize our contribution by stating that the “only” substantial difference between our method and qDARTS is the lack of Gumbel sampling. We made a deliberate design decision to remove Gumbel sampling because it introduces non-physical quantum states. Therefore porting DARTS while maintaining physical interpretability necessitated a complete overhaul of the QAS framework. Although the reviewer claims that this is merely an incremental contribution, the density matrix formalism of DARTS and the noise-aware search setting, that they themselves have acknowledged as strengths, were only made possible due to this change.
>
> Lastly, the reviewer points out how the scalability issue of simulation-based QAS approaches is the limiting factor in their applicability. We agree with this statement and further argue that due to the exponential scaling it is unrealistic to expect simulation-based QAS methods to discover full circuits for large-scale problems. Instead, we believe there should be a shift in how QAS experiments are designed, with a focus on discovering reusable quantum subroutines that generalize across qubit counts for the same problem class, and we briefly discuss this in appendix A.
>
> However, as such an experimental paradigm is not commonly pursued in QAS literature, our experiments in this work are designed explicitly to draw comparison to existing QAS methods. The results of our comparison show that although the density matrix simulations require more space, $O(4^n)$, they enable us to save on runtime when compared to sampling-based QAS with statevector simulations because we need to run fewer simulations in total. As evidence, we provide the following table comparing the average wall times and iteration times for the entangled state generation experiments. Note that a majority of the qDARTS runs did not converge on an optimal circuit and that many RhoDARTS runs terminated early.
> |Entanglement Type|Search Algorithm|Num Qubits|Wall Time (min)|Simulation Time (s)|
> |-|-|-|-|-|
> |GHZ|qDARTS|2|1.21|0.023|
> |||3|1.76|0.032|
> |||4|3.71|0.040|
> |||5|8.02|0.048|
> |||6|9.54|0.057|
> ||$\rho$DARTS|2|0.17|0.018|
> |||3|0.21|0.023|
> |||4|0.27|0.028|
> |||5|0.74|0.037|
> |||6|0.49|0.046|
> |W|qDARTS|2|10.96|0.085|
> |||3|20.13|0.121|
> |||4|25.37|0.152|
> |||5|41.95|0.252|
> |||6|63.81|0.383|
> ||$\rho$DARTS|2|0.40|0.039|
> |||3|0.71|0.089|
> |||4|29.59|0.178|
> |||5|48.94|0.294|
> |||6|68.14|0.409|

---

### Official Review · Reviewer_BCZt · 2025-11-01

**Soundness:** 2
**Presentation:** 2
**Contribution:** 1
**Rating:** 2
**Confidence:** 5

**Summary:**

The author extended the qDARTS which is doing differentiable architecture search over noiseless unitary circuits, to search over density matrix. The author perform different experiments to validate their methods.

**Strengths:**

1. Th $\rho$-based search might be useful for noise-aware design
2. The author show extensive experiments
3. The paper is well written, and in good structure

**Weaknesses:**

1. The proposed method does not provide clear advantages. For the experiments in the paper, it is generally equivalent to searching with state vectors. Circuits are unitary and objectives are linear in the state. The gradients are equivalent: $\partial_\theta f=\partial_\theta\langle\psi| O|\psi\rangle=2 \operatorname{Re}\left\langle\partial_\theta \psi\right| O|\psi\rangle=\operatorname{Tr}\left[O \partial_\theta \rho\right]$.
2. A state vector stores $2^n$ complex amplitudes, whereas a density matrix stores a $2^n\times 2^n$ complex array. Memory and time therefore scale like $O(2^n)$ for state vectors and $O(4^n)$ for density matrices, an extra factor of $2^n$.
3. Training the architecture is a classical differentiable program that does not require a quantum device. It is unclear whether this approach can find circuits that are classically hard to simulate.
4. The experimental results do not show definitive advantages over other methods, making it even harder to justify the extra factor of $2^n$.

**Questions:**

1. Can the authors comment on the scalability of their method?
2. Can the authors clearly state the advantages of their method over others?

---

> ### Author Response · Authors · 2025-11-20
>
> We thank the reviewer for their feedback.
>
> We would first like to address the reviewer’s claim that the search setting is equivalent to using statevector simulations. The gradient equality provided is only true if the density matrix represents a pure state, $\rho = \ket\psi\bra\psi$, whereas our method makes use of mixed states. In particular the quantum operations that update the quantum state, defined in equations 4 and 6, are not unitary circuits. For this reason, it is incorrect to say that our search setting is equivalent to a search using statevector simulations.
>
> In fact, the design decision to use mixed states is what enables our method’s main advantage: faster convergence. Quantum simulation, with its exponential overhead, is the most expensive operation of simulation-based QAS. The mixed state formalism allows us to bypass sampling and hence converge with fewer quantum simulations when compared to sampling-based QAS; empirical evidence of this is discussed in lines 373-377. The practical advantage of needing fewer total simulations is observing a shorter run time for convergence.
>
> The reviewer also claims that the experimental results do not show definitive advantages over the benchmarks, which is untrue. For tasks I and III, the results reported in Fig. 3, Table 2 and Fig. 5 very clearly show our method to outperform the qDARTS benchmark. For task II, our results in Table 1 show that our method generates comparable results to the benchmark and as discussed in lines 373-377, it does so while requiring far fewer quantum simulations.
>
> The reviewer questions whether our approach can find circuits that are classically hard to simulate because our method is a classical algorithm. It is our understanding that a quantum circuit that can be efficiently simulated must either exploit the symmetries of the underlying problem, e.g. with tensor-network approximations, or use a restricted gate set e.g. Clifford gates. However, our method does not use  information of any underlying structure that may be present in the VQA problem, and it allows for arbitrary gate sets to construct the circuits. That is to say, our method does not have an inherent bias to find circuits that can be efficiently simulated with classical computers.
>
> Lastly, the reviewer has asked us to comment on the scalability of our method. We direct the reviewer to appendix D, which discusses the computational complexity of our method. As mentioned before, all simulation-based QAS algorithms scale exponentially due to the overhead of simulation. We also direct the reviewer to appendix A where we briefly discuss how simulation-based QAS could find better applicability in spite of their exponential scaling by designing experiments to find reusable quantum circuits.

---

> > ### Comment · Reviewer_BCZt · 2025-11-20
> >
> > I would like to thank the authors for their detailed answers.
> >
> > First, I agree that the gradient expression is equivalent when the state $\rho$ is pure. However, for the molecular Hamiltonian and Max Cut problems, and the usual state initializations, the target is a pure ground state. In these settings your method can be viewed as optimizing an ensemble of circuits that approximate the target, rather than a single circuit architecture. This makes the terminology of “circuit architecture search” confusing.
> >
> > I also appreciate the references to Fig. 3, Table 2 and Fig. 5. However, these comparisons are all made only against qDARTS. It remains unclear what advantage your approach has over other methods for variational circuit design and architecture search such as reinforcement learning based approaches, or ADAPT-VQE type methods.
> >
> > Concerning potential implementation on quantum hardware, the current formulation requires propagating the full mixed state $\rho$, which is not physically realizable as a single process on a device. In the formalism of Eq. (6) and the combinatorial nature of the space, it is not clear how this would scale in terms of the number of required runs on a quantum device. I would like to ask the authors to comment explicitly on scalability.
> >
> > If one restricts to classical devices, then exact density matrix based propagation is asymptotically more expensive than state vector simulation. In that regime it is not clear for which problem sizes or circuit families the proposed method would be advantageous. The approach appears difficult to scale beyond relatively small system sizes.

---

> > > ### Author Response · Authors · 2025-11-28
> > >
> > > We thank the reviewer for their continued engagement in this discussion, and we address their concerns below.
> > >
> > > ## On the terminology “architecture search”
> > > We agree that in our experiments, we employed our method to optimize an ensemble of circuits to approximate a single target state. We do see an ambiguity in interpreting the term “architecture search” in this context as the generated architectures are not subsequently reused to solve other instances of problem, e.g. different molecular geometries in VQE. We recognize that the applicability of QAS is limited when focusing on such single circuit experiments, and have briefly discussed this in appendix A.
> > >
> > > At the same time, we follow this experimental framework because it is an established practice in the QAS literature. For example, the qDARTS VQE experiments considered only one molecular configuration for each molecule. The CRLQAS authors reused the same molecular configurations to provide results that are directly comparable to the previous works in QAS. More broadly, other QAS approaches, including ADAPT-VQE, also tend to construct new ansatze for each problem instance. Therefore, to ensure a fair comparison against existing QAS baselines our experiments focus on per-instance architecture discovery, consistent with how the term “architecture search” is used in the field today.
> > >
> > >
> > > ## Advantages over other QAS approaches
> > > Sampling-based QAS methods define probability distributions to select circuit architectures. By sampling circuits from this probabilistic ensemble and evaluating the states they produce with the loss function, the selection probabilities are updated to find optimal architectures that solve the underlying problem. Some methods like qDARTS and DQAS explicitly define the selection probabilities and directly optimize them. On the other hand RL methods, which generate policies that stochastically construct architectures, implicitly define the selection probabilities in the weights of the RL agent.
> > >
> > > Our method follows the explicit definition of selection probabilities, but removes the sampling operation which provides a global view of the search space. The practical advantage of this is seen in the convergence time. As discussed in lines 373-377, our method converges with far fewer simulations and therefore takes less time to find optimal circuits. As a concrete example, for the LiH-4 benchmark, the authors of CRLQAS report an episode time of ~25 seconds with 15,000 episodes, i.e. a wall time of 104 hours. In contrast our LiH-4 run reported in Table 1 terminated early, converging after 1,095 simulations, taking 18.65 minutes.
> > >
> > > Because our method shares an optimization strategy with sampling-based QAS methods, we can make direct numerical comparisons which demonstrate the effects of not sampling circuits. However, for other QAS approaches with different optimization/search strategies, e.g. ADAPT-VQE, such a direct comparison may not be possible.
> > >
> > > ## On hardware feasibility and scalability
> > > The reviewer’s concerns on porting our method to quantum hardware is well founded. To our knowledge, implementing the quantum operation $\mathcal E$ directly on hardware is not feasible without sampling individual circuits from the search space. This reintroduces the sampling overhead that our method aims to eliminate, and would also make the gradients ill-defined. For this reason, our method is solely a simulation-based QAS approach.
> > >
> > > With respect to the scalability of the quantum simulation, the reviewer correctly points out that the density matrix propagation is asymptotically more expensive than statevector simulation. However, statevector simulators are limited to simple noise models such as bit/phase flips and readout error. All quantum simulations that allow generic noise modelling will scale as $O(4^n)$, and as such, our method’s scalability is not worse than any other simulation-based QAS approach that accounts for noise.

---

### Official Review · Reviewer_FUzL · 2025-11-01

**Soundness:** 2
**Presentation:** 3
**Contribution:** 2
**Rating:** 4
**Confidence:** 3

**Summary:**

This paper proposes a novel quantum architecture search algorithm based on the mixed-state formalism, where the search process is interpreted as the evolution of mixed quantum states. The mixture arises from randomness in the distribution over potential circuit elements. The authors benchmark their method and demonstrate that it outperforms the baseline, qDART, both with and without noise.

**Strengths:**

- The paper introduces a new differentiable quantum architecture search algorithm that is conceptually simple and elegant.


- Numerical results show strong performance improvements over qDART, and the algorithm exhibits notable robustness to noise, particularly in the noise probability range of 0.01–0.1.

**Weaknesses:**

- The experiments are conducted only in simulation, not on real quantum hardware. As a result, it remains unclear how ρDART would perform under more realistic noise conditions. Since the algorithm requires exponential time when simulated classically, its practical utility relies on implementation on real quantum hardware. Demonstrating strong performance under current NISQ hardware would therefore be a crucial validation.

- Moreover, when executed on real hardware, the method would likely lose its “sampling-free” advantage, as multiple experimental shots would be required to estimate the loss. It is also unclear how gradients would be computed in this setting.

- While the approach is technically sound, the conceptual advance over qDART appears limited, the key difference being the substitution of Gumbel-softmax with softmax. The contribution would be more compelling if supported by stronger and more rigorous experimental results.

**Questions:**

Can the proposed algorithm be extended to run efficiently on real quantum hardware, thereby avoiding the exponential simulation cost?

---

> ### Author Response · Authors · 2025-11-20
>
> We thank the reviewer for their feedback and comments.
>
> We would first like to address the conceptual advantage of our approach to qDARTS. In particular, qDARTS makes use of the Gumbel softmax operation which generates outputs that are not interpretable as physical quantum statevectors. In designing $\rho$DARTS, we wanted to avoid introducing non-physical states. To adopt the DARTS framework while maintaining physical interpretability, we introduced mixed states which are generated from probabilistic ensembles of the circuits in the search space. Unlike qDARTS which samples from the search space, $\rho$DARTS considers the entire search space simultaneously. Empirical results show that our method converges to optimal circuits after a fewer number of simulations when compared to sampling based QAS methods like qDARTS that explicitly samples circuits, and RL agents that implicitly sample circuits.
>
> The reviewer has correctly pointed out, our experiments are not conducted on quantum computing hardware, and that attempting to do so would eliminate the sampling-free nature of our algorithm, and the gradients would not be well defined. However, the method that we have proposed is not a quantum algorithm, the search can only run on a classical computer. The reason for this is that the quantum operation $\mathcal{E}$ defined in equations 4 and 6 can only be implemented on hardware by sampling from the gate distributions. The classical nature of our QAS approach is not unique, in fact the non-physicalities introduced by Gumbel sampling also render qDARTS to be a purely classical algorithm.

---

> > ### Comment · Reviewer_FUzL · 2025-11-26
> > **Thank you for the response**
> >
> > Thank you for the responses. I understand the classical nature of quantum architecture search (QAS) algorithms: since their output is a variational quantum circuit (VQC), the QAS procedure itself does not necessarily need to be a quantum algorithm.
> >
> > However, this classicality raises concerns about scalability, as QAS requires expensive quantum circuit simulations that do not scale well on classical hardware. For a truly scalable QAS method, it is desirable for the resource requirements to grow only polynomially, which in practice suggests the need for a quantum algorithm. I would be very surprised if a classical polynomial-time QAS algorithm exists, though I agree that exploring this direction is worthwhile. While I recognize that competing works also face scalability issues, this does not lower the bar; rather, I view it as a major gap in the literature that remains important to address.
> >
> > On the other hand, even if classical QAS is acceptable, a direct way to evaluate a QAS method is to test the resulting VQC on real hardware. This aspect is missing from the current paper. Real devices often involve more complex and device-specific noise characteristics than those considered in classical simulations. If the VQC produced by your approach performs well on actual quantum hardware (i.e., if it generalizes across different noise models) that would be a strong advantage.
> >
> > With this in mind, I would like to maintain my current score.

---

> > > ### Author Response · Authors · 2025-11-28
> > >
> > > We appreciate the reviewer’s careful consideration of our work. Even if the reviewer chooses to maintain their current assessment, we hope that the clarifications below will provide further context about our experiments and the scope of our contributions.
> > >
> > > ## Scalability of classical QAS
> > > We agree with the reviewer’s assessment that a polynomial-time classical QAS is unlikely to exist in general, and this is largely due to the exponential overhead required in simulating quantum systems. Further, if the QAS method is able to find circuits for general problem families, i.e. the QAS method does not exploit the unique properties of any one individual problem, and explores a search space of general circuits which may not be efficiently simulable, exponential scaling is expected.
> > >
> > > Within this context, our goal is not to claim a breakthrough in asymptotic scaling, but to contribute a method that is competitive in the regime of simulation-based QAS. Importantly, our density-matrix formulation has the same asymptotic scaling as other QAS methods that support generic noise modeling. Thus, while we share the fundamental limitations inherent to classical simulation, our method does not introduce additional scalability penalties beyond existing noisy simulations, and further has an advantage in convergence time when compared to sampling-based simulated QAS methods.
> > >
> > > ## Hardware validation
> > > We appreciate the reviewer’s point that, to evaluate QAS methods, validating the generated circuits on hardware is a valuable complementary test. We likewise agree that demonstrating the robustness of the discovered circuits across different hardware noise models would be a particularly meaningful advantage. Our simulation framework already supports general noise channels, making it possible to incorporate device-specific noise models if the target hardware and its relevant calibration data are known.
> > >
> > > However, we believe that adding device-specific noise to simulations without also adopting the native gate set of the device would not yield meaningful insight. The simulated QAS literature typically constructs circuits from a universal gate set (usually CNOT gates and Pauli rotations) at a higher level of abstraction than the physical gate set of the quantum computer. Because the noise characteristics post transpilation can differ substantially from those of the abstract circuit, incorporating hardware-level noise while retaining a non-native gate set would likely misrepresent actual performance.
> > >
> > > Exploring this direction is important, and our framework is compatible with such extensions. However, in this work, our goal was to follow the established benchmarking protocols in the QAS literature, which evaluate architectures through simulation under configurable, device-agnostic noise models to enable fair comparison between different QAS approaches.

---

### Official Review · Reviewer_GY2Y · 2025-11-06

**Soundness:** 2
**Presentation:** 3
**Contribution:** 3
**Rating:** 4
**Confidence:** 4

**Summary:**

This paper presents a novel differentiable quantum architecture search (QAS) algorithm by modeling the search process as the evolution of a mixed quantum state. Compared to previous QAS algorithms, this new method has two prominent features: (1) it is "sample-free", in the sense that no quantum circuits need to be sampled in the learning process; (2) the mixed state formulation naturally enables the method to incorporate generic noise models, such as depolarizing channels, which is not supported in previous state-vector-based simulation. Numerical experiments on VQE and QAOA demonstrate that this new method requires significantly fewer quantum simulations, with an improved level of noise resilience.

**Strengths:**

- The formulation based on mixed states is novel in the literature of differentiable QAS, and it facilitates a more efficient exploration of the search space by eliminating the need to sample quantum circuits.
- This framework allows the modeling of hardware noise, which is crucial for the implementation of VQAs on near-term quantum devices.
- Numerical experiments show good performance for a range of standard tasks, including (entangled) state preparation, quantum chemistry (VQE), and Max-Cut problems.

**Weaknesses:**

- While the numerical experiments show promising performance in terms of energy errors or state fidelity, the runtime of this new $\rho$DARTS algorithm is not reported. In particular, an end-to-end efficiency measure (e.g., wall-clock time) does not appear to be taken into account. Therefore, it is hard to judge the practical value of this new method.
- The scalability of this proposed method has not been fully discussed either. If this type of differentiable QAS can only be simulated using classical devices, it can not be scaled up to a few tens of qubits. However, even moderate chemistry/optimization problems may involve up to hundreds of qubits. At this scale, is this method still applicable?

**Questions:**

- How is the runtime of $\rho$DARTS compared to existing methods? Is it going to be significantly more time-consuming (to achieve comparable results)?
- Is it possible for this new differentiable QAS algorithm to generate better circuits for error mitigation (i.e., enabling an active search for noise-resilient variational circuits, instead of passive noise robustness)?
- In principle, can we implement a similar differentiable QAS using a quantum computer (for better scalability)? If not, what are the potential barriers?

---

> ### Author Response · Authors · 2025-11-20
>
> We thank the reviewer for feedback, and we address their concerns below.
>
> We provide the following table containing the average wall times of our entangled state generation experiments, along with the mean runtime of the individual quantum simulations:
> |Entanglement Type | Search Algorithm | Num Qubits | Wall Time (min)| Simulation Time (s) |
> |-|-|-|-|-|
> |GHZ | qDARTS      | 2 |  1.21 | 0.023 |
> |    |             | 3 |  1.76 | 0.032 |
> |    |             | 4 |  3.71 | 0.040 |
> |    |             | 5 |  8.02 | 0.048 |
> |    |             | 6 |  9.54 | 0.057 |
> |    | $\rho$DARTS | 2 |  0.17 | 0.018 |
> |    |             | 3 |  0.21 | 0.023 |
> |    |             | 4 |  0.27 | 0.028 |
> |    |             | 5 |  0.74 | 0.037 |
> |    |             | 6 |  0.49 | 0.046 |
> |W   | qDARTS      | 2 | 10.96 | 0.085 |
> |    |             | 3 | 20.13 | 0.121 |
> |    |             | 4 | 25.37 | 0.152 |
> |    |             | 5 | 41.95 | 0.252 |
> |    |             | 6 | 63.81 | 0.383 |
> |    | $\rho$DARTS | 2 |  0.40 | 0.039 |
> |    |             | 3 |  0.71 | 0.089 |
> |    |             | 4 | 29.59 | 0.178 |
> |    |             | 5 | 48.94 | 0.294 |
> |    |             | 6 | 68.14 | 0.409 |
>
> We remind the reviewer that in these experiments, the maximum number of simulations was kept constant for both search methods. Further, a majority of the qDARTS runs were unable to find an adequate circuit under these conditions, and many of the rhoDARTS runs converged before terminating early. While the simulation time for rhoDARTS may exceed that of qDARTS, rhoDARTS converges to an optimal quantum circuit with far fewer simulations.
>
> We are unable to make a direct comparison of the wall times for the VQE experiments as we benchmarked against the results reported in the qDARTS and CRLQAS papers. While the qDARTS authors do not specify any run times, the CRLQAS authors report an episode run time of ~25 seconds for the LiH-4 molecule in the noiseless simulation for a gate count of 40. It is unclear whether this is equivalent to the search settings they used in the main paper, where they ran 15,000 episodes per run. If so, their wall time would have been 104 hours, where they first reach chemical accuracy after half of that time. In comparison, our LiH-4 run provided in Table 1 terminated early after 1,095 simulations, with a wall time of 18.65 minutes.
>
> Regarding the scalability of our method, the space and time complexity are discussed in appendix D. The reviewer correctly points out that simulation-based QAS methods have a scalability ceiling that depends on the number of qubits that can be simulated on classical hardware. In its present form, our method is not applicable to simulate several hundreds of qubits which may be needed for chemistry problems, though this is also true of the statevector simulators.
>
> To answer the question about the error mitigation, yes, our method allows one to actively search for variational circuits that perform well in the presence of the simulated noise model.
>
> Lastly, we do not currently know how to port our method to run on a quantum computer. The biggest barrier to do so is implementing the operations $\mathcal{E}$ on hardware while maintaining our sampling-free QAS approach.

---

### Author Response · Authors · 2025-12-02
**Summary of the discussion 1**

We thank all the reviewers for their responses and participation in the discussion. We summarize the key strengths highlighted by the reviewers, the major concerns raised and our clarifications provided in the rebuttal below.
# Strengths
The following strengths were consistently identified by the reviewers:
- **Theoretical framework**\
Multiple reviewers highlighted that the mixed state formulation provides an elegant and physically sound translation of the classical DARTS framework to a quantum setting that eliminates the sampling cost of other QAS methods.
- **Native noise-aware setting**\
Our use of density matrix simulations enabled us to model generic quantum noise channels which are relevant in finding circuits for modern quantum systems. Several reviewers regarded the method’s robustness as a strong result of our paper, especially compared to methods based on statevector simulations.
- **Extensive numerical experiments**\
Reviewers found our experiments demonstrated our method to outperform the qDARTS benchmark on a wide range of standard VQA tasks.

# Concerns and Clarifications
## Incremental contribution
Reviewer 51Km asserted that the novelty of our work is limited due to the experimental framework being similar to qDARTS. The similarity arises due to the shared foundation of classical DARTS which inspired both algorithms, hence motivating a direct comparison between the two. We emphasized that removing Gumbel sampling, the core contribution of our method, was not a superficial change; its motivation was to incorporate physical interpretability, which is notably absent in qDARTS. This necessitated a complete overhaul of the simulation framework to use mixed states which consequently enabled the sampling-free and noise-aware simulations that were identified as strengths by the same reviewer.

## Ambiguity of the term “architecture search”
Reviewer BCZt noted that the use of “architecture search” may appear ambiguous given that search is performed per problem instance rather than for reusable architectures. We clarified that this per-instance formulation follows established practice in the QAS literature, including qDARTS, CRLQAS, and ADAPT-VQE, and allows direct comparison with prior work.

## Run time
Reviewer GY2Y requested clarification on practical run-time performance. We provided wall-clock statistics showing that RhoDARTS converges faster than qDARTS in entangled-state preparation, and that its convergence time for VQE is orders of magnitude smaller than CRLQAS when extrapolating their reported episode durations. We directly attribute the shorter convergence time to the sampling-free nature of our method.

## Discretization gap
Reviewer 51Km brought up the discretization gap present in DARTS-like methods, stating that there is no guarantee that the pure state produced by the final, discrete architecture would match the optimized performance of the mixed state defined by the continuous search parameters. We showed that the distribution of loss values for each pure state in the ensemble is centered around the loss value of the mixed state, and that because we choose the architecture with the highest probability, we expect the final loss value to be close to that of the mixed state. Further, we also introduced NME regularization which penalizes high entropy gate distributions, thereby eliminating the discretization gap.

## Scalability
While the reviewers tend to agree that an exponential overhead is unavoidable, especially if the search is not biased to find efficiently simulable circuits, their primary concern was that density matrix propagation scales quadratically worse than a statevector simulation. However, statevector simulations can only model simple noise channels such as bit/phase flips and readout errors; they are unable to capture generic noise models relevant to NISQ systems. Any quantum simulation that accounts for generic noise will inevitably require density matrices which makes our method’s scaling equivalent to other noise-aware, simulated QAS methods. Furthermore, in the noiseless regime, we’ve shown that the sampling-free nature of our method enables a practical advantage in run time even when compared to sampling-based QAS using statevector simulations.

## Porting to hardware
The reviewers questioned if our method could, in principle, be ported to run on quantum hardware. We clarified that our approach is based on simulation because the quantum operation $\mathcal E$ cannot be implemented on hardware without sampling circuits from the search space, which our method was designed to avoid.

---

> ### Author Response · Authors · 2025-12-02
> **Summary of the discussion 2**
>
> ## Hardware validation
> Reviewer FUzL raised the question of validating our discovered circuits on real quantum hardware and assessing their robustness to unseen, device-specific noise. In our response, we clarified that while our simulation framework can incorporate device-specific noise, the experiments in the paper intentionally follow the established benchmarking conventions in simulated QAS, which use a universal gate set and device-agnostic noise models to enable fair comparison. We also noted that introducing hardware-specific noise without simultaneously adopting the device’s native gate set would not give meaningful insight, as the noise model used in training would not correspond to noise affecting the transpiled circuit. Our method is compatible with extensions toward hardware-targeted studies, but without access to a hardware platform, our focus is to provide accurate comparison to existing QAS methods.
>
> ## Advantages over other QAS
> Reviewer BCZt questioned the advantages of our method in comparison to other QAS methods, particularly reinforcement learning and ADAPT-VQE. While we cannot articulate direct advantages over ADAPT-VQE, as it is fundamentally different in structure from our method, we can clearly observe the advantages from eliminating the sampling cost from sampling-based QAS methods, including other differentiable QAS and RL-based approaches. We find that RhoDARTS requires substantially fewer simulations to converge which yields a practical runtime advantage.
>
> # Conclusion
> We appreciate the time and care the reviewers devoted to evaluating our work. While it is unfortunate that the discussion period was cut short, we hope that the clarifications provided in our responses offer the AC a clear and accurate picture of the paper’s contributions, its positioning within the QAS literature, and the intent behind our experimental design choices. We believe the additional context we have supplied meaningfully addresses the reviewers’ concerns and highlights the strengths and practical value of our approach.

---

### Meta-Review · Area_Chair_HFjs · 2025-12-23

**Summary:**

This paper proposes ρDARTS, a differentiable quantum architecture search (QAS) method that reformulates the DARTS relaxation using quantum mixed-state dynamics. By representing the architecture search space as a density matrix, the method avoids circuit sampling and enables gradient-based optimization over the entire ensemble of candidate circuits. The authors evaluate the approach on state preparation, VQE, and Max-Cut tasks, showing faster convergence (fewer simulations) and improved robustness to noise compared to qDARTS, albeit within a classical simulation setting.

All reviewers provided negative evaluations. Multiple reviewers raised serious concerns about scalability, incremental contribution relative to qDARTS, and, most importantly, the lack of deployability on quantum hardware. Several reviewers also questioned the authors’ claim that QAS is inherently a classical procedure and expressed concern that the proposed method is fundamentally restricted to small-scale classical simulations due to the O(4^n) cost of density-matrix evolution.

Given the above issues, I therefore recommend rejection.

**Reviewer Concerns:**

**Concerns Addressed by the Rebuttal**

The rebuttal provided additional clarification and evidence on runtime behavior in simulation, including wall-clock timing comparisons showing faster convergence than qDARTS under equal simulation budgets. The authors also clarified the conceptual distinction between mixed-state evolution and statevector-based search, and addressed the discretization gap by introducing entropy-based regularization to encourage collapse toward a single architecture.

**Concerns Remaining After the Rebuttal**
1. *Fundamental lack of deployability on quantum hardware*. The authors explicitly acknowledge that their method cannot be run on quantum hardware in principle, because the sampling-free mixed-state evolution and differentiable updates cannot be implemented without sampling individual circuits, which would destroy the core mechanism of the method. This is a critical limitation, as many existing QAS approaches (e.g., neural QAS) are naturally deployable on quantum devices. As a result, it is not applicable to large circuits or realistic quantum systems, severely limiting its practical relevance.

2. *Incorrect claim that QAS is inherently a classical algorithm*. The rebuttal repeatedly asserts that quantum architecture search is fundamentally a classical procedure. This claim is incorrect: many QAS methods are explicitly designed to run on quantum hardware via circuit sampling and measurement-based evaluation.

3. *Limited comparative scope*. Empirical comparisons are restricted almost exclusively to qDARTS. Concerns about how the method compares to other QAS paradigms (e.g., RL-based approaches or ADAPT-VQE) remain largely unaddressed.

**Reviewer Scores:**

Given that the rebuttal confirms fundamental limitations of the proposed method, it is unlikely that any reviewer would have increased their score after participating fully in the discussion. In particular, clarifications regarding scalability and hardware infeasibility would likely have reinforced the concerns of more critical reviewers.

---

### Decision · Program_Chairs · 2026-01-26

Reject